# ZEROTH-ORDER FORWARD-ONLY SNN TRAINING INSPIRING NEUROMORPHIC ON-CHIP LEARNING

## ABSTRACT

The human brain is a biologically instantiated on-device neural system that integrates both learning and inference in a unified architecture, which enables rapid and flexible learning on-the-fly. This extraordinary capability is achieved through non-backpropagation learning mechanisms, whereas backpropagation (BP) is computationally and memory intensive which makes it unsuitable for on-chip edge learning. Zeroth-order (ZO) optimization methods, which resemble biologically plausible perturbation-based learning, offer a promising alternative that enables learning with only forward passes and hence can significantly reduce the complexity of on-chip hardware implementation. However, in this work we show that applying ZO methods to spiking neural networks (SNNs) is non-trivial due to the step-function nature of spiking activation (e.g., Heaviside function). We analyze the challenges posed by the step-function activation, and propose a novel subspace-based zeroth-order (SZO) learning method that leverages the intrinsic low-dimensional structure of the SNN optimization trajectory. By learning in a low-dimensional subspace, SZO substantially enhances ZO learning efficacy, achieving accuracy comparable to first-order (FO) methods with faster learning speed than full-space BP. We evaluate SZO on model training from scratch, continual training, and unsupervised adaptation. Experimental results demonstrate that SZO closely approaches FO training performance for the first time while offering fast learning speed. We expect this work to inspire future research on highly efficient and scalable algorithms for neuromorphic on-chip learning. Code is available at `https://anonymous.4open.science/r/SZO-2B08`.

## 1 INTRODUCTION

For edge AI applications, two brain-like capabilities are particularly desirable: *1) High energy efficiency. 2) Extraordinary online learning ability.* Functionally, the brain can be regarded as a biologically instantiated on-device neural system that integrates both learning and inference within a unified architecture, which enables flexible and adaptive learning on-the-fly. Realizing these capabilities in neuromorphic computing systems to achieve efficient on-chip learning for edge devices has great promise (Christensen et al., 2022), which has inspired many researches on spiking neural networks (SNNs) (Maass, 1997; Zenke & Ganguli, 2018; Lee et al., 2016; Neftci et al., 2019; Fang et al., 2021), and neuromorphic hardware that support on-chip learning, such as BrainScaleS (Schemmel et al., 2010) and Loihi (Davies et al., 2018). A key challenge lies in designing highly efficient SNN learning algorithms with low hardware implementation complexity.

Recent years have witnessed rapid progress in SNN algorithms that successfully narrows the performance gap with ANNs (Meng et al., 2023; Hu et al., 2021; Li et al., 2021; Yin et al., 2021; Xiao et al., 2022; Fang et al., 2023b; Zhou et al., 2022; Yao et al., 2024; Zhou et al., 2024). Most of these advances rely on BPTT. However, BPTT requires unfolding the backward computational graph across time-steps, which incurs substantial computational and memory overhead that makes it unsuitable for resource-constrained edge devices. To mitigate this, approximate BPTT algorithms using only forward-in-time computation are proposed (Bellec et al., 2020; Xiao et al., 2022; Meng et al., 2023; Yin et al., 2023). Nonetheless, these algorithms still rely on spatial BP, which remains too demanding for practical on-chip implementation, especially for modest and large models.

Specifically, BP is not only biologically implausible (Lillicrap et al., 2020) but also too complex to implement on resource-constrained edge hardware: **(1) the need for dedicated feedback circuits** for error backpropagation, **(2) the update locking problem**, where forward computation must wait for backward computation, **(3) high memory overhead** from storing activations, especially as batch-size and time-steps increase, **(4) inefficient weight transport**, involving hardware-inefficient weight transfer during backward passes. In comparison, local learning approaches (Kaiser et al., 2020) and Hebbian learning methods are highly parallelizable in hardware and have been successfully implemented on neuromorphic platforms (Furber et al., 2014; Davies et al., 2018; Baek et al., 2019; Frenkel et al., 2018; 2019; Heidarpur et al., 2019). Nevertheless, these methods lack scalability to deep models and complex tasks.

Zeroth-order (ZO) optimization, which only requires forward computations, offers an attractive alternative for neuromorphic hardware due to its simplicity. Recent studies have shown its effectiveness for fine-tuning LLMs (Malladi et al., 2023; Zhang et al., 2024). From a bio-inspired perspective, perturbation-based learning, a long-studied biologically grounded approach (Fiete et al., 2007; Bouvier et al., 2018; Aronov et al., 2008; Ali et al., 2013), is an instantiation of ZO optimization. Since it avoids backward gradient computation, it is well suited for hardware implementation without feedback paths and holds great promise for efficient neuromorphic on-chip learning.

However, despite this potential, we show that applying ZO methods to train SNNs is non-trivial due to the step-function nature of the spiking activation (e.g., Heaviside function). Specifically, the step function leads to high variance in ZO gradient estimation, which makes ZO methods ineffective for SNN training. To address this, we propose a subspace-based ZO (SZO) learning algorithm that leverages the intrinsic low-dimensional structure of SNN optimization trajectory to achieve effective SNNs training using forward-only passes. The main contributions are summarized as follows.

- We show that the step-function activation in SNNs introduces high variance in ZO gradient estimation, which substantially degrades the effectiveness of ZO methods.
- We propose a subspace-based zeroth-order (SZO) learning algorithm, which can achieve accuracy comparable to first-order (FO) methods with faster learning speed than vanilla (full-space) BP, which is particularly desirable for on-chip learning scenarios.
- We provide theoretical analysis to show that the proposed method can improve the convergence rate and stability of ZO training.
- We conduct extensive evaluations of SZO across three learning scenarios, including training from scratch, continual training, and unsupervised adaptation, which demonstrates that SZO achieves state-of-the-art accuracy on these tasks, closely approaching the performance of FO training.

To the best of our knowledge, SZO is the first forward-passes-only SNN training method that can scale to large-scale datasets such as ImageNet. In typical on-device learning settings, training is performed upon pretrained models. This setup echoes the brain's learning paradigm (Lillicrap et al., 2020), where neural architectures and synapse strengths are innately initialized and only modest learning is required to acquire new tasks. Therefore, our method offers a promising step toward enabling efficient, scalable neuromorphic on-chip learning for real-world applications.

It is worth noting that Mukhoty et al. (2023) employs local ZO differentiation at the neuron level to address SNN non-differentiability, but still depends on global BP. Moreover, Xiao et al. (2024) proposes a pseudo-ZO algorithm for SNN based on node perturbation in forward computation, yet it requires error feedback to each layer by feedback connections, which resembles direct feedback alignment (Nøkland, 2016). A detailed related work is provided in Appendix A.

## 2 PRELIMINARIES

### 2.1 SPIKING NEURAL NETWORKS

Inspired by the brain, SNNs operate in a spike-based processing paradigm that transmits information via temporally sparse binary spikes. This enables energy-efficient processing well-suited to neuromorphic hardware. A widely used neuron model in SNNs is the leaky integrate-and-fire (LIF) model (Abbott, 1999; Burkitt, 2006), whose discrete-time dynamics can be expressed as

$$u_l[t] = \left(1 - \frac{1}{\tau}\right)(u_l[t-1] - V_{th}s_l[t-1]) + W_l s_{l-1}[t] + b_l, \tag{1}$$

$$s_l[t] = H\left(u_l[t] - V_{th}\right) = \begin{cases} 1, & u_l[t] > V_{th} \\ 0, & \text{otherwise} \end{cases}, \tag{2}$$

where $u_l[t]$ and $s_l[t]$ denote the membrane potential and output spike vector at time step $t$ for layer $l$, respectively. $W_l$ and $b_l$ are the corresponding synaptic weights and bias. $V_{th}$ is the spike emission threshold. A spike is emitted when the membrane potential exceeds $V_{th}$, after which the potential is reset by subtracting $V_{th}$. $H\left(\cdot\right)$ is the Heaviside activation function, which is non-differentiable and poses challenges for gradient-based training. To address this, surrogate gradient methods have been widely used to enable effective training of SNNs via BPTT (Neftci et al., 2019).

## 2.2 ZEROTH-ORDER OPTIMIZATION

ZO optimization refers to a class of optimization methods that rely solely on function evaluations without requiring access to explicit gradient information. These methods are particularly well-suited for scenarios where the objective function is non-differentiable or its gradient is inaccessible or costly to compute. Two widely used ZO gradient estimation approaches are coordinate-wise gradient estimation (CGE) and randomized vector-wise gradient estimation (RGE), defined as

$$\text{(CGE)} \qquad \hat{\nabla}^{\text{CGE}} \mathcal{L}_{\mathcal{B}}\left(\theta\right) = \sum_{i=1}^{d} \frac{\mathcal{L}_{\mathcal{B}}\left(\theta + \mu e_i\right) - \mathcal{L}_{\mathcal{B}}\left(\theta - \mu e_i\right)}{2\mu} e_i, \tag{3}$$

$$\text{(RGE)} \qquad \hat{\nabla}^{\text{RGE}} \mathcal{L}_{\mathcal{B}}\left(\theta\right) = \frac{1}{q} \sum_{i=1}^{q} \frac{\mathcal{L}_{\mathcal{B}}\left(\theta + \mu z_i\right) - \mathcal{L}_{\mathcal{B}}\left(\theta - \mu z_i\right)}{2\mu} z_i, \quad z_i \sim \mathcal{N}(0, I_d), \tag{4}$$

where $\theta \in \mathbb{R}^d$ denotes the model parameters to be optimized. The function $\mathcal{L}_{\mathcal{B}}(\theta) := \frac{1}{B} \sum_{i=1}^{B} \mathcal{L}_{x_i}(\theta)$ is the empirical loss evaluated over a mini-batch $\mathcal{B} = \{x_1, \ldots, x_B\}$ of size $B$, with each $x_i$ sampled independently from the distribution $\mathcal{D}$. Here, $\mathcal{L}_x(\theta)$ is the loss evaluated at instance $x$, and the expected loss is defined as $\mathcal{L}(\theta) = \mathbb{E}_x[\mathcal{L}_x(\theta)] = \mathbb{E}_{\mathcal{B}}[\mathcal{L}_{\mathcal{B}}(\theta)]$. $\mu > 0$ controls the magnitude of parameter perturbations. In CGE, $e_i \in \mathbb{R}^d$ corresponds to the $i$-th standard basis vector, where all entries are zero except the $i$-th component equal to 1. For RGE, $z_i \in \mathbb{R}^d$ is a random vector sampled from isotropic distributions, e.g., standard Gaussian $\mathcal{N}(0, I)$ or uniform distribution. The computational complexity of CGE scales linearly with $d$ due to its coordinate-wise operations. In contrast, RGE uses $q$ random perturbations for each gradient estimation. CGE can achieve much better performance than RGE when optimizing deep neural models (Chen et al., 2024a).

## 3 ANALYSIS ON THE VARIANCE AMPLIFICATION OF HEAVISIDE ACTIVATION

While ZO optimization has recently demonstrated success in training ANNs (Chen et al., 2024a) and fine-tuning LLMs (Malladi et al., 2023), it faces significant challenges when training SNNs. Figure 1a compares the performance of ZO training of ANN and SNN on CIFAR10 using the same ResNet20 architecture (Chen et al., 2024a). The SNN variant is derived by replacing ReLU activations in the ANN ResNet20 with LIF neuron (1), with a time-step of 1. Both networks are trained for 40 epochs using the DeepZero method with its default hyperparameter setting, which employs CGE (3) and has demonstrated success in training deep ANNs while RGE (4) fails (Chen et al., 2024a). The results in Figure 1a, together with additional intensive experiments, reveal that ZO methods fail to train SNN. In contrast, DeepZero with CGE performs well on ANN. This highlight a limitation of applying ZO methods directly to SNNs.

To understand the underlying cause of this limitation, we analyze the variance of ZO gradient estimation in ANN and SNN training. Given estimated gradients $G = [\hat{g}_1, \hat{g}_2, \ldots, \hat{g}_K] \in \mathbb{R}^{d \times K}$ across $K$ mini-batches with mean gradient $\bar{g} = \frac{1}{K} \sum_{k=1}^{K} \hat{g}_k$, we compute the variance as $\text{Var}(\hat{g}) = \frac{1}{K} \sum_{k=1}^{K} \|\hat{g}_k - \bar{g}\|^2$. The left sub-figure in Figure 1b compares the gradient estimation variance across different layers of the ANN and SNN. It can be seen that, in the early training stage, the variance of the shallow and middle layers of the SNN is much higher than that of the ANN. Moreover, the right sub-figure in Figure 1b compares the proportion of effective perturbations. A perturbation $e_i$ is considered effective if $\mathcal{L}_{\mathcal{B}}(\theta + \mu e_i) \neq \mathcal{L}_{\mathcal{B}}(\theta)$. Clearly, the SNN has a much lower ratio of effective perturbations compared to the ANN.

To further probe the origin of the high variance of SNN, we compare the variance induced by different activation functions, including ReLU, Sigmoid, and Heaviside.

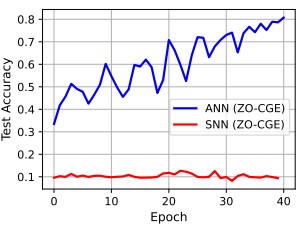
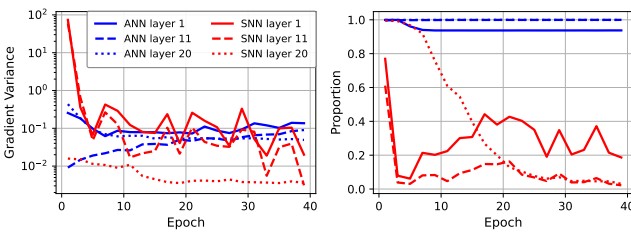

(a) Test accuracy curves      (b) Gradient variance and proportion of effective perturbations

Figure 1: Comparison of ZO training of ANN and SNN using DeepZero with CGE.

**Proposition 1** (Variance comparison between activate functions). *Let $z$ be a random variable with variance $\sigma^2$ and satisfying that $z$ is symmetric about the origin, i.e. $\Pr(z > t) = \Pr(z < -t)$, and $z$ has a continuous density in a neighbourhood of $0$ such that $\Pr(z = 0) = 0$. Define $f_h(z) := \mathbb{I}(z > 0)$, $f_s(z) := \frac{1}{1+e^{-z}}$, and $f_r(z) := \max\{0, z\}$. Then there exists a sufficient small $\sigma_0 > 0$ such that for any $\sigma \in (0, \sigma_0)$ there holds: $\mathrm{Var}\big(f_h(z)\big) > \mathrm{Var}\big(f_s(z)\big)$ and $\mathrm{Var}\big(f_h(z)\big) > \mathrm{Var}\big(f_r(z)\big)$.*

This result indicates that the Heaviside activation introduces significantly higher output variance for a small input perturbation. For a Gaussian example $z \sim \mathcal{N}(0, \sigma^2)$, we have $\mathrm{Var}(f_h(z)) = 0.25$, $\mathrm{Var}(f_s(z)) \approx 0.0625\sigma^2$, and $\mathrm{Var}(f_r(z)) \approx 0.341\sigma^2$. Thus, for a small $\sigma$ we have $\mathrm{Var}(f_h(z)) \gg \mathrm{Var}(f_r(z)) > \mathrm{Var}(f_s(z))$. In SNNs, this high variance leads to unstable training dynamics and rapid ill-conditioning after only a few epochs, as shown in Figure 1b. Simultaneously, the proportion of effective perturbations diminishes drastically, which leads to training failure. These findings motivate the use of specialized variance reduction approaches in ZO optimization for SNNs.

## 4 THE PROPOSED SZO METHOD

### 4.1 LOW-DIMENSIONAL STRUCTURE IN SNN TRAINING

It has been well shown that the training trajectory of ANNs can be approximately covered by a small $k$-dimension space with $k \ll d$ (Li et al., 2018; Larsen et al., 2021; Gur-Ari et al., 2018; Gressmann et al., 2020; Li et al., 2022). Here we examine this property for SNNs to verify that the low-dimension trajectory hypothesis is also true for SNNs. We conduct experiments using Spike-ResNet20 (0.27M) and Spike-ResNet18 (11.22M) on CIFAR10 and CIFAR100, respectively. The training is performed using BPTT with surrogate gradient. We set the time-step to 4. This training achieve a test accuracy of 86.65% on CIFAR10 and 74.42% on CIFAR100.

The models are trained for 150 epochs. In the training procedure, we sample $N = 150$ model weights $\{\theta_1, \theta_2..., \theta_N\}$ after each epoch and perform centered PCA. Specifically, the sampled weights are first centralized as $\Theta = [\theta_1 - \bar{\theta}, \theta_1 - \bar{\theta}, ..., \theta_N - \bar{\theta}]$, where $\bar{\theta} = \frac{1}{N}\sum_{i=1}^{N}\theta_i$. Then, we conduct spectral decomposition on $\Theta\Theta^\top$ and obtain $k$ eigenvectors $P = [v_1, v_2, ..., v_k]$ corresponding to the top-$k$ largest eigenvalues. As depicted

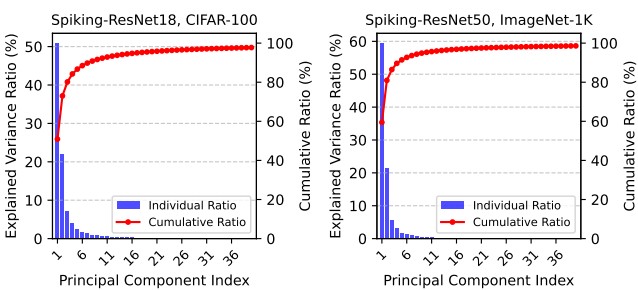

Figure 2: Explained variance analysis of the sampled weights $\Theta$.

in Figure 2, The first ten components account for more than 90% of the total variance for both Spike-ResNet18 on CIFAR100 and Spike-ResNet50 on ImageNet1K. This indicates the existence of a low-dimensional principal subspace that can cover the dominant dynamics of training.

Moreover, as shown in Figure 3 in experiments, training within a 60-dimension subspace spanned by the top eigenvectors $P$ can yield test accuracy comparable to full parameter space training. These results confirm that SNN training trajectories also exhibit a low-dimension structure, and deep SNNs can be efficiently trained in a low-dimension principal subspace. Notably, as shown in Figure 3, subspace-based training converges significantly faster than full-space BP. A theoretical justification of the improved convergence rate of subspace-based learning is provided in Section 5.

### 4.2 SUBSPACE-BASED ZEROTH-ORDER OPTIMIZATION FOR SNN

The variance of ZO gradient estimation scales linearly with the parameter dimension $d$, which hinders its applicability to deep networks. For example, the RGE (4) has a variance in the order $O(d/q)$ (Duchi et al., 2015), where $q$ is the query number. In SNNs this problem is further aggravated by the variance amplification effect of Heaviside activation, as shown in Section 3. To address this challenge, we exploit the low-dimension structure of SNN optimization to substantially reduce the variance of ZO gradient estimation and improve the performance of ZO-based SNN training.

Specifically, given the subspace projection matrix $P = [v_1, v_2, ..., v_k] \in \mathbb{R}^{d \times k}$ with $k \ll d$ as described in Section 4.1, we optimize an SNN in the subspace spanned by $P$ by CGE or RGE as

$$
\begin{aligned}
\text{(SZO-CGE)} \quad & \hat{\nabla}^{\text{SZO}} \mathcal{L}_{\mathcal{B}}(\theta) = \sum_{i=1}^{k} \frac{\mathcal{L}_{\mathcal{B}}(\theta + \mu v_i) - \mathcal{L}_{\mathcal{B}}(\theta - \mu v_i)}{2\mu} v_i, \\
\text{(SZO-RGE)} \quad & \hat{\nabla}^{\text{SZO}} \mathcal{L}_{\mathcal{B}}(\theta) = \frac{1}{q} \sum_{i=1}^{q} \frac{\mathcal{L}_{\mathcal{B}}(\theta + \mu P z_i) - \mathcal{L}_{\mathcal{B}}(\theta - \mu P z_i)}{2\mu} P z_i, \quad z_i \sim \mathcal{N}(0, I_k).
\end{aligned}
\tag{5}
$$

The parameter update rule with momentum is given by

$$
\theta_{t+1} = \theta_t - \eta \hat{\nabla}^{\text{SZO}} \mathcal{L}_{\mathcal{B}_t}(\theta_t) + \beta(\theta_t - \theta_{t-1}),
\tag{6}
$$

where $\eta$ is the learning rate and $\beta \in [0, 1)$ is the momentum coefficient. SZO-CGE involves $2k$ forward passes to estimate the gradient. In experiments, $k$ is typically on the order of a few tens of dimensions, which is much smaller than the parameter dimension $d$, i.e., $k \ll d$. So SZO-CGE can retain the advantage of CGE but without incurring significant computational overhead. We also propose SZO-RGE, which uses $q$ random perturbations sampled within the subspace to estimate the gradient. According to our experiments, SZO-RGE can further reduce forward passes while achieving comparable accuracy in most cases. Detailed sketch of SZO is summarized in Appendix B.

In this work, we employ PCA to obtain subspace, as it provides a principled method to rank and select basis vectors according to their importance, thereby reducing the number of bases and minimizing memory requirements. While other orthonormalization techniques, such as the Gram-Schmidt process, could also be used, the specific choice of basis construction method is not central to our work. The core of our idea is to construct the subspace from the model's training trajectory. Thus, the fundamental contribution remains unchanged regardless of the specific tool used to form the orthonormal basis.

## 5 THEORETICAL ANALYSIS

We provide analysis to show that subspace-based ZO training improves both convergence and stability compared with full-space ZO training. Proofs are given in Appendix. We begin by introducing some basic assumptions. The first is the standard $L$-smoothness assumption on the loss, which is commonly required in convergence analysis of non-convex neural networks. Since SNNs are non-differentiable and even generalized Clarke differential (which requires continuity) does not exist for spiking (Heaviside) activations, currently no existing mathematical tool can directly analyze their convergence rate. Given that our primary goal is to highlight the benefits of subspace-based ZO learning in a general non-convex setting, using the $L$-smoothness assumption is a reasonable and sufficient choice for this.

**Assumption 1** (Basic Assumptions). *We impose the following fundamental assumptions on the loss function $\mathcal{L}$ and the projection matrix $P$:*

1. *The loss function $\mathcal{L}_x(\theta)$ is L-smooth with respect to $\theta$, and its gradient norm is uniformly bounded, i.e., $\|\nabla \mathcal{L}_x(\theta)\| \leq G$ for all $\theta \in \mathbb{R}^d$. This also implies $\|\nabla \mathcal{L}(\theta)\| \leq G$.*

2. *For the projection matrix $P$, throughout the training process of $T$ steps, there exists $0 < \varepsilon \leq \frac{1-\beta}{L\sqrt{T}}$ such that $\|PP^{\top}\nabla \mathcal{L}_x(\theta_t) - \nabla \mathcal{L}_x(\theta_t)\| \leq \varepsilon$ for all training step $t$ and $x \sim \mathcal{D}$. Consequently, $\|PP^{\top}\nabla \mathcal{L}_{\mathcal{B}}(\theta_t) - \nabla \mathcal{L}_{\mathcal{B}}(\theta_t)\| \leq \varepsilon$ also holds.*

3. *The variance due to sampling $x$ is bounded by $\sigma^2$ for any $\theta$, i.e., $\mathbb{E}_x[\|\nabla \mathcal{L}_x(\theta) - \nabla \mathcal{L}(\theta)\|^2] \leq \sigma^2$, which further implies $\mathbb{E}_{\mathcal{B}}[\|\nabla \mathcal{L}_{\mathcal{B}}(\theta) - \nabla \mathcal{L}(\theta)\|^2] \leq \sigma^2/B$.*

The second assumption ensures that the gradient can be projected into the subspace and subsequently recovered with only a small discrepancy. This aligns with our empirical results, which indicate that the optimization trajectory of neural networks in classification tasks predominantly lies within a subspace, and the projection matrix captures the basis of this subspace. The third assumption is a common condition in the analysis of mini-batch SGD algorithms.

To facilitate the analysis, we further define $p_t = \frac{\beta}{1-\beta}(\theta_t - \theta_{t-1})$ and $m_t = \theta_t + p_t$. This yields the following relation:

$$\theta_{t+1} + p_{t+1} = \theta_t + p_t - \frac{\eta}{1-\beta}\hat{\nabla}^{\text{SZO}}\mathcal{L}_{\mathcal{B}_t}(\theta) \Rightarrow m_{t+1} = m_t - \frac{\eta}{1-\beta}\hat{\nabla}^{\text{SZO}}\mathcal{L}_{\mathcal{B}}(\theta).$$

To establish the convergence rate of the algorithm, we first derive a bound on the difference between the estimated gradient and the true gradient based on a mini-batch data.

**Lemma 2** (Estimation error of the SZO gradient estimator). *Suppose the loss function $\mathcal{L}$ and the projection matrix $P$ satisfy Assumption 1, and let $\mathcal{B}$ be a mini-batch of size $B$. Then, under the gradient estimator and parameter update rule given in (5) and (6), for any parameter $\theta_t$, the following bound holds:*

$$\|\hat{\nabla}^{SZO}\mathcal{L}_{\mathcal{B}}(\theta_t) - \nabla\mathcal{L}_{\mathcal{B}}(\theta_t)\|^2 \leq \frac{1}{2}k\mu^2 L^2 \|P\|_{op}^4 + 2\varepsilon^2,$$

*where $\|\cdot\|_{op}$ denotes the operator norm.*

Lemma 2 demonstrates that the discrepancy in the SZO gradient estimator is governed by the perturbation magnitude $\mu$ and the representational capacity of the projection matrix, quantified by $\varepsilon$. Leveraging this lemma, we subsequently derive the overall convergence rate.

**Theorem 3** (Convergence rate of SZO). *Suppose the loss function $\mathcal{L}$ and the projection matrix $P$ satisfy Assumption 1. Consider minimizing $\mathcal{L}$ using mini-batch gradients over $T$ training steps, with momentum parameter $\beta$, learning rate $\eta = \frac{1-\beta}{L\sqrt{T}}$, and perturbation scale $\mu \leq \frac{1-\beta}{L\sqrt{T}}$. Under the update rule (6), the convergence rate of the algorithm satisfies*

$$\frac{1}{T}\sum_{t=0}^{T-1}\mathbb{E}\left[\|\nabla\mathcal{L}(\theta_t)\|^2\right] \leq \mathcal{O}\left(\frac{k(1-\beta)^2}{\sqrt{T}} + \left(\frac{\beta}{1-\beta}\right)^2 \frac{\sigma^2}{B\sqrt{T}}\right).$$

Theorem 3 demonstrates that subspace-based ZO achieves a convergence rate of $\mathcal{O}(k/\sqrt{T})$, which offers a substantial improvement over the full-space ZO optimization rate of $\mathcal{O}(d/\sqrt{T})$, as $k \ll d$.

As our main focus is to employ the SZO method to continual train or adapt a pretrained model, the following result further demonstrates improved local stability of subspace based ZO optimization.

**Theorem 4** (Local Stability of SZO and CGE). *Let $\theta^*$ be a critical point of the expected loss function, i.e., $\nabla\mathcal{L}(\theta^*) = 0$. Suppose the batch-normalized covariance $\Sigma$ satisfies $\Sigma \preceq \sigma^2 I_d/B$, where $\sigma^2$ is the variance specified in Assumption 1. Define $\bar{P} = PP^\top$. Then, local mean-square stability at $\theta^*$ for both CGE and SZO with mini-batch SGD is ensured if*

$$(\nabla\mathcal{L}(\theta))^\top(\theta - \theta^*) > \frac{\eta}{2}\text{Tr}(\Sigma), \quad \text{and} \quad (\nabla\mathcal{L}(\theta))^\top\bar{P}(\theta - \theta^*) > \frac{\eta}{2}\text{Tr}(\bar{P}\Sigma\bar{P}^\top),$$

*respectively. Furthermore, if there exists a neighborhood $\mathcal{V}(\theta^*)$ such that local strong monotonicity holds with constant $\rho$, then local mean-square stability at $\theta^*$ for CGE and SZO is guaranteed if $\eta < \frac{2B\rho r^2}{d\sigma^2}$ and $\eta < \frac{2B\rho r^2}{k\sigma^2}$, respectively, where $r := \sup_{\theta\in\mathcal{V}(\theta^*)}\|\bar{P}(\theta - \theta^*)\|$.*

Theorem 4 shows that subspace-based ZO training can largely relax the condition for local stability, as $\text{Tr}(\bar{P}\Sigma\bar{P}^\top) \ll \text{Tr}(\Sigma)$ when $k \ll d$. For example, the step-size for stability can be relaxed from $\eta < \mathcal{O}(1/d)$ to $\eta < \mathcal{O}(1/k)$. This indicating SZO can be more stable than CGE.

## 6  EXPERIMENTS

We evaluate the proposed method on three tasks: model training from scratch, continual training, and unsupervised adaptation. We consider four SNN models, Spike-ResNet20 (0.27M), Spike-ResNet18

(11.2M), Spike-ResNet50 (25.56M) (Hu et al., 2024), and Spike-VGG11 (9.5M), which are trained and tested on CIFAR10, CIFAR100, ImageNet1K and DVS-Gesture datasets, respectively. We take 4 time-steps for static datasets and 20 time-steps for DVS-Gesture. In SZO, we use a perturbation scale $\mu = 1$, a momentum coefficient $\beta = 0.9$, without weight decay. The learning rate is decayed using a cosine method, with an initial value of 1. We compare SZO with full-space BP and subspace-based BP, which are trained by BPTT with surrogate gradient. Moreover, subspace-based BP and SZO use the same subspace. For subspace-based BP, the parameters are updated using the BP gradient projected into the subspace. We employ the TET loss function (Deng et al., 2022) for all supervised training ($\lambda = 0.05$, $\phi = 1$), and the entropy loss (Wang et al., 2020) for unsupervised model adaptation, maintaining identical running statistics of all BN layers for all perturbations in one time step. More training details gives in Appendix G. We use SpikingJelly (Fang et al., 2023a) for SNN training and all experiments are conducted on eight NVIDIA V100 GPUs, each with 32GB of memory.

## 6.1 TRAINING FROM SCRATCH IN SUBSPACE

For CIFAR10 and CIFAR100, the model sampling number is $N = 150$, and the subspace dimension is $k = 60$. For DVS-Gesture, the model sampling number is $N = 80$, and the subspace dimension $k = 20$. The models are sampled every epoch from the full-space BP training. For ImageNet1K, we sample $N = 150$ models from epoch 0 to 60 uniformly and extract a subspace dimension of $k = 60$.

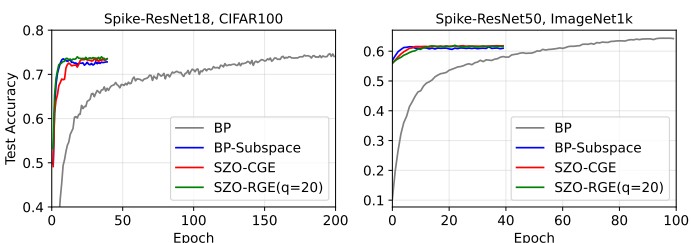

Figure 3: Training dynamics of different methods.

SZO and subspace-based BP are trained for 40 epochs on all datasets, while BP is trained for 100 epochs on ImageNet1K and 200 epochs on other datasets to converge. Table 1 and Figure 3 show that SZO can achieve comparable performance with that of BP and subspace-based BP. In particular, subspace-based training converges significantly faster than full-space BP. For instance, on ImageNet-1K, training in the subspace for fewer than 10 epochs can achieve the similar accuracy as 60 epochs of full-space BP. This rapid learning capability is particularly desirable for on-chip learning scenarios.

CGE typically outperforms RGE in high-dimensional settings such as Chen et al. (2024a). However, SZO operates in a very low-dimensional subspace (tens), which largely eliminates the difference between SZO-CGE and SZO-RGE.

Table 1: Accuracy (%) comparison on training from scratch

| Dataset | Model | Params | BP | BP-Subspace | SZO-CGE | SZO-RGE($q=1$) | SZO-RGE($q=20$) |
|---------|-------|--------|-----|-------------|---------|----------------|-----------------|
| CIFAR10 | Spike-ResNet20 | 0.27M | 86.74±0.12 | 86.42±0.32 | 86.21±0.17 | 81.29±0.35 | 83.71±0.34 |
| CIFAR100 | Spike-ResNet18 | 11.22M | 74.51±0.16 | 73.32±0.22 | 73.35±0.30 | 73.77±0.26 | 73.79±0.22 |
| DVS-Gesture | Spike-VGG11 | 9.50M | 97.57±0.35 | 96.65±1.06 | 96.65±0.40 | 91.78±1.64 | 95.72±0.40 |
| ImageNet1K | Spike-ResNet50 | 25.56M | 63.81±0.78 | 61.71±0.30 | 61.71±0.18 | 61.82±0.10 | 61.92±0.08 |

## 6.2 MODEL CONTINUAL TRAINING

To verify that the extracted subspace enhances ZO continual training, we first pre-train the Spike-ResNet18 model with full-space BP for 100 epochs, sampling from epochs 80 to 100 to create a 20-dimensional subspace for CIFAR100 using PCA. For ImageNet1K, we pre-train Spike-ResNet50 for 60 epochs and extract a 20-dimensional subspace by sampling from epoch

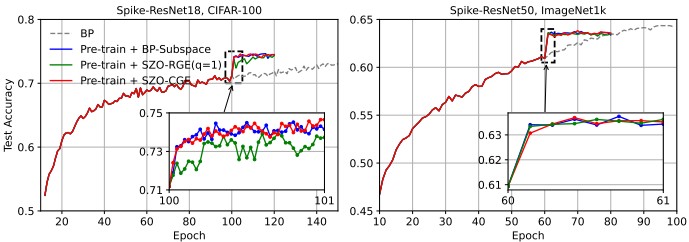

Figure 4: Test accuracy of the compared methods on continual training task.

Table 2: Accuracy (%) comparison on unsupervised model adaptation on CIFAR10C, CIFAR100C and ImageNetC (all with severity level 5).

| Setting | Dataset | Model | No adapt. | BN | Tent (BP) | BP-Subspace | SZO-CGE | SZO-RGE ($q=10$) |
|---|---|---|---|---|---|---|---|---|
| Reset | CIFAR10C | Spike-ResNet20 | 45.01 | 65.53 | 66.14 | 67.69 | 67.09 | 57.09 |
| | CIFAR100C | Spike-ResNet18 | 29.81 | 51.72 | 52.54 | 75.02 | 72.47 | 63.03 |
| | ImageNetC | Spike-ResNet50 | 9.36 | 21.20 | 22.82 | 31.76 | 32.94 | 32.34 |
| Continual | CIFAR10C | Spike-ResNet20 | 45.01 | 65.53 | 64.58 | 72.73 | 76.10 | 31.09 |
| | CIFAR100C | Spike-ResNet18 | 29.81 | 51.72 | 53.84 | 78.15 | 79.69 | 75.86 |
| | ImageNetC | Spike-ResNet50 | 9.36 | 21.20 | 3.89 | 32.15 | 33.37 | 33.16 |

50 to 60. Continual training is then performed using subspace-based methods for 20 epochs based on the pre-trained model. Figure 4 compares the continual training performance of SZO with BP.

It can be seen that, after switching to subspace-based training, SZO exhibits a sharp increase in accuracy within a single epoch on both CIFAR100 and ImageNet1K. On CIFAR100, SZO achieves an accuracy exceeding 74% in just one epoch, which closely approaches the converged accuracy of full-space BP after approximately 50 epochs. This demonstrates the significant acceleration enabled by subspace-based learning. The improved convergence rate is theoretically justified in Section 5.

Finally, on CIFAR100 SZO-CGE continual training achieves an accuracy of 74.59%, which is even higher than that of full-space BP (74.42%). On ImageNet1K, SZO-CGE achieves an accuracy of 63.73%, approaching the converged accuracy of full-space BP (64.36%). SZO-RGE achieves comparable maximum accuracy to SZO-CGE, but requires less training time, as detailed in Table 3.

## 6.3 Unsupervised Model Adaptation

We further consider unsupervised model adaptation, especially test-time adaptation (TTA) (Schneider et al., 2020; Wang et al., 2020), which adapts a source model to a target domain using only test data in an unsupervised manner. We evaluate pretrained Spike-ResNet20, Spike-ResNet18 and Spike-ResNet50 models on CIFAR10C, CIFAR100C and ImageNetC, respectively. CIFAR100C and ImageNetC each contain 15 corruption types. Models are adapted over a single cycle of the 15 corruptions. In this cross-domain setting, while using the subspace extracted from source pretraining can also significantly improve cross-domain adaptation performance, relying on it alone remains insufficient to achieve optimal results comparable to state-of-the-art methods. Thus, we construct a subspace specialized for corruption adaptation. Rather than building one for each corruption, we find that using a representative corruption suffices for generalization. Specifically, for ImageNetC, we select Gaussian (severity 5), and sample 20 BP-trained models in 20 supervised training epochs. Then, these models are combined with 20 source models to extract the subspace with 20 dimensions using PCA. For CIFAR10C and CIFAR100C, we only sample 20 models trained on Gaussian (severity 1) in 10 supervised training epochs, combing with 20 source models and reduce dimension to 20 using PCA. We evaluate under *Reset* and *Continual* settings for each method. The *Reset* setting means the model is reset to its initial parameters after adapting to one corruption type, while the *Continual* setting means the model continuously adapts to the 15 corruption types without resetting.

Table 2 compares SZO with two representative TTA methods, BN (Schneider et al., 2020) and Tent (Wang et al., 2020). More results are provided in Appendix E. SZO achieves significant better performance than Tent trained with full-space BP. The better performance of SZO is attributed to the stability provided by training in low-dimensional subspace, which mitigates the instability of using unsupervised entropy loss as entropy minimization may be misled by samples with high confidence (i.e., low entropy) but incorrect predictions. This advantage of subspace training is more obvious under the *Continual* setting. These results demonstrate that SZO is effective when the subspace is extracted from source tasks that share similarity with the downstream tasks.

## 6.4 Computational Efficiency

**GPU runtime memory.** GPU memory of BP, BP with mixed-precision (BP-Mix), SZO and ZO without subspace, recorded via `torch.cuda.max_memory_allocated()` are shown in Figure 5. As BP and BP-Mix must store all inter-layer activations, its memory usage grows steeply with both batch-size and time-steps. In contrast, SZO does not store these activations, thus its memory usage increases only marginally as batch-size increases and remains almost invariant to time-step. Although SZO need to store subspace projection matrix, this overhead is significantly smaller than the activation storage required by BP and BP-mix.

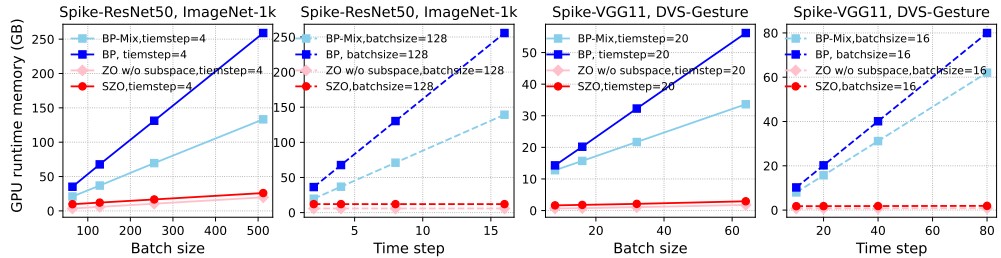

Figure 5: GPU runtime memory under varying batch sizes and time steps.

Table 3: Comparison of training time in the continual training and training from scratch task. #FP and #BP denote the number of forward and backward passes. For CIFAR100, all methods run on a single V100 GPU. For ImageNet1K, BP and BP-subspace use 8 V100s, while SZO-CGE/RGE use 2 V100s.

| Dataset / Model | Method | Continual training | | | | | Training from scratch | | | | |
|---|---|---|---|---|---|---|---|---|---|---|---|
| | | Converged epoch | Converged time (min) | #FP | #BP | Max Acc(%) | Converged epoch | Converged time (min) | #FP | #BP | Max Acc(%) |
| CIFAR100 / Spike-ResNet18 | BP | 97 | 58.2 | 30107 | 30107 | 74.42 | 197 | 142.8 | 77027 | 77027 | 74.42 |
| | BP-Subspace | 4 | 3.4 | 1564 | 1564 | 74.64 | 8 | 9.4 | 3128 | 3128 | 73.37 |
| | SZO-CGE | 4 | 69.6 | 64124 | 0 | 74.59 | 38 | 1335.2 | 1797818 | 0 | 73.42 |
| | SZO-RGE($q=1$) | 6 | 8.7 | 7038 | 0 | 74.42 | 39 | 60.7 | 45747 | 0 | 73.91 |
| ImageNet1K / Spike-ResNet50 | BP | 35 | 1248.5 | 41316 | 41316 | 64.36 | 95 | 3939.6 | 118940 | 118940 | 64.36 |
| | BP-Subspace | 1 | 41.6 | 1252 | 1252 | 63.74 | 7 | 324.8 | 8764 | 8764 | 61.49 |
| | SZO-CGE | 3 | 1104.3 | 78876 | 0 | 63.73 | 18 | 32011.2 | 2726856 | 0 | 61.84 |
| | SZO-RGE($q=1$) | 1 | 56.5 | 3756 | 0 | 63.62 | 28 | 1484.6 | 105168 | 0 | 61.78 |

**Training time.** Table 3 compares the training time to reach maximum accuracy in the continual training and training from scratch tasks. The training time for continual training is measured starting from the beginning of the continual training stage. Although SZO involves more forward computations per iteration, it converges faster and thus reaches maximum accuracy in fewer epochs. Moreover, we found that SZO-RGE further reduces the number of forward passes while achieving comparable accuracy in most cases.

## 6.5 ABLATION STUDY

**Perturbation scale.** Table 4 shows the accuracy of SZO-CGE in the training from scratch experiment for different perturbation scales. It performs well on a wide range of perturbation scales.

Table 4: Accuracy (%) of SZO-CGE on training from scratch task with different perturbation scales.

| Dataset | Network | $\mu = 0.6$ | $\mu = 1.0$ | $\mu = 1.4$ | $\mu = 1.8$ | $\mu = 2.2$ |
|---|---|---|---|---|---|---|
| CIFAR10 | Spike-ResNet20 | 85.91±0.31 | 86.21±0.17 | 86.01±0.41 | 86.03±0.29 | 85.92±0.12 |
| CIFAR100 | Spike-ResNet18 | 73.20±0.33 | 73.35±0.30 | 73.09±0.41 | 73.07±0.24 | 72.98±0.47 |

**Model sampling window size and subspace dimension.** In constructing the subspace, the models are typically sampled once per epoch starting from the initial weight. A smaller window size $N$ means that fewer source-pretraining epochs are needed, while a lower subspace dimension $k$ results in less memory overhead. Figure 6 shows that SZO-CGE using a larger subspace dimension can achieve higher accuracy in training from scratch task.



Figure 6: Accuracy of SZO-CGE with different sample window size and subspace dimension.

## 7 DISCUSSION AND CONCLUSION

**SZO: Enabling efficient neuromorphic on-chip learning.** SZO is motivated to overcome the barriers that make BP unsuitable for hardware deployment: the need for dedicated feedback circuits, the update-locking problem, high memory overhead, and inefficient weight transport, as detailed in Section 1. In contrast, SZO relies only on forward computation and offers strong suitability for practical on-chip learning: **(1)** *No dedicated feedback circuits are required.* **(2)** Learning in a low-dimensional subspace enables significantly *faster convergence* compared to full-space BP, which

is particularly desirable for real-time on-chip learning. **(3)** Although SZO requires storing a subspace matrix of size $d \times k$, it eliminates the need to store activations, thus has *far lower memory overhead than BP* (Figure 5). **(4)** Unlike vanilla ZO methods that require repeatedly sampling high-dimensional random vectors of size $d$, SZO only samples within a subspace of dimension $k$ (typically tens, e.g., $k = 20, 60$ vs. $d = 11.22$M and $25.56$M for Spike-ResNet18 and Spike-ResNet50). This dimensionality reduction not only accelerates optimization but also *removes the hardware bottleneck of generating and manipulating extremely high-dimensional random perturbations*, which makes ZO learning substantially more hardware-efficient and practically deployable on neuromorphic chips.

**Conclusion.** In this work, we revealed the underlying cause of the ineffectiveness ZO optimization on training SNNs, and proposed a subspace-based ZO method for SNN training. Further, we provided theoretical analysis to show that subspace-based ZO training can improve the convergence and stability compared with the full-space ZO training. Experiments on model training from scratch, continual training, and unsupervised adaptation demonstrated that SZO not only closely approaches FO training performance but also offering much faster learning speed, which is particularly desirable for on-chip learning. SZO is the first forward-passes-only SNN training method that can scale to large-scale datasets such as ImageNet1K. Exploring learning-based approaches (e.g., meta-learning over similar source tasks) for subspace construction can be expected to further improve performance on downstream tasks, which is an important direction for future work.

## REPRODUCIBILITY STATEMENT

We are committed to ensuring the reproducibility of our results. Codes for reproducing all the results in our paper are available at `https://anonymous.4open.science/r/SZO-2B08`. Details of our experimental setup, including model architectures, data preprocessing, and hyperparameter settings, are provided in Section 6 and Appendix G. Extensive results are provided in Appendix C and E.

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

# A    RELATED WORK

**SNN Learning Methods.** The high energy efficiency of SNNs has inspired extensive researches (Maass, 1997; Zenke & Ganguli, 2018; Lee et al., 2016; Rueckauer et al., 2017; Wu et al., 2018; Neftci et al., 2019; Lee et al., 2020a; Fang et al., 2021; Radhakrishnan et al., 2021; Zheng et al., 2021; Han et al., 2020; Rathi & Roy, 2021). A central challenge in training large SNN models lies in the non-differentiability of spiking activations. Early approaches using ANN-to-SNN conversion have demonstrated promising performance on large models by leveraging pre-trained artificial neural networks (Hu et al., 2023; Wu et al., 2021; Pérez-Carrasco et al., 2013; Han et al., 2020; Bu et al., 2023). More recently, a growing body of work has focused on directly training using surrogate gradients with BPTT, which successfully narrow the performance gap between SNNs and ANNs on complex tasks and larger models (Meng et al., 2023; Hu et al., 2021; Li et al., 2021; Yin et al., 2021; Xiao et al., 2022; Zhang & Zhang, 2024; Fang et al., 2023b; Zhou et al., 2022; Yao et al., 2024; Zhou et al., 2024). To address the high complexity of BPTT, approximate methods have been proposed (Bellec et al., 2020; Bohnstingl et al., 2022; Xiao et al., 2022; Meng et al., 2023; Yin et al., 2023).

**Biologically Plausible Methods.** BP is considered biologically implausible due to its weight transport and update locking problems (Lillicrap et al., 2020). Feedback alignment (Lillicrap et al., 2016) and its variants (Nøkland, 2016; Launay et al., 2020; Frenkel et al., 2021; Liao et al., 2016; Xiao et al., 2019; Akrout et al., 2019) have been proposed to relax the requirement of exact weight symmetry. Target propagation (Lee et al., 2015) provides an alternative by propagating local targets instead of gradients. Hebbian learning has long been viewed as a biologically grounded mechanism for synaptic plasticity (von der Malsburg, 1973; Kohonen, 1982; Oja, 1982; Song & Abbott, 2001; Markram et al., 1997; Bi & Poo, 1998; Gu, 2002; Magee & Grienberger, 2020; Krotov & Hopfield, 2019), which updates synaptic weights based solely on local pre- and post-synaptic activity. Perturbation-based methods offer another biologically plausible approach by estimating learning signals through perturbations (Mazzoni et al., 1991; Seung, 2003; London et al., 2010; Fiete et al., 2007; Miconi, 2017; Bouvier et al., 2018; Aronov et al., 2008; Ali et al., 2013; Flower & Jabri, 1992; Hiratani et al., 2022), which directly send loss differences back to weights and thus do not require any backward passes and weights. However, these methods typically face scalability challenges when applied to large models or complex tasks.

**BP-Free Methods for Forward-Only Learning.** To avoid the problems of BP, such as update-locking and high memory requirement, BP-free methods, such as the forward gradient (FG) and ZO optimization methods, have recently been shown to be utilizable for neural network training. The FG methods compute gradients based on the directional derivatives that can be computed efficiently via forward differentiation mode (Baydin et al., 2022; Ren et al., 2023; Bacho & Chu, 2024; Fournier et al., 2023). ZO methods not only avoid the backward gradient computation but also can largely reduce the memory overhead (Chen et al., 2024a). Particularly, it has been recently shown that ZO optimization can effectively finetune large language models (Malladi et al., 2023; Zhang et al., 2024; Guo et al., 2025).

**Subspace-Based Learning.** The low-dimensional structure underlying neural network optimization has been well recognized (Li et al., 2018; Larsen et al., 2021; Gur-Ari et al., 2018; Gressmann et al., 2020; Li et al., 2022), which show that optimization trajectories often reside within intrinsic low-dimensional subspaces. Moreover, recent approaches have exploited low-rank structures to enable efficient fine-tuning of large models through low-rank representation (Hu et al., 2022).

**Subspace-based BP-Free Methods in LLM finetuning.** Current applications of subspace methods combined with ZO optimization are predominantly found in the finetuning of LLMs. For instance, while (Malladi et al., 2023) did not propose a subspace-based ZO optimization algorithm, their theoretical analysis under the Polyak-Łojasiewicz (PL) condition derived a convergence rate related to the effective rank. Several other works, including (Jin et al., 2024; Chen et al., 2024c; Yu et al., 2024; 2025), have combined ZO methods with subspace techniques like LoRA, using random projections to construct the subspace ZO approach. Specifically, (Yu et al., 2024) established a convergence rate for a plain ZO-SGD method based on random projection, assuming a convex loss function. In the non-convex case, (Chen et al., 2024c) provided a convergence rate for their random projection-based plain ZO-SGD as $O(\sqrt{\frac{d}{rT}})$, where $d$ is the number of parameters, $r$ is the subspace dimension, and $T$ is the training time. Additionally, (Yu et al., 2025) obtained a convergence rate of $O(\frac{d}{T})$ in non-convex setting.

**Neuromorphic On-Chip Learning Hardware.** Recent years have seen growing interest in neuromorphic on-chip learning. Early mixed-signal chips, such as BrainScaleS (Schemmel et al., 2010; Billaudelle et al., 2020) and ROLLS (Qiao et al., 2015), use STDP-like mechanisms for local synaptic updates. Digital neuromorphic systems, such as SpiNNaker (Furber et al., 2014), Loihi (Davies et al., 2018), and many others (Baek et al., 2019; Frenkel et al., 2018; 2019; Heidarpur et al., 2019), support programmable plasticity and on-chip learning based on STDP with hardware-efficient designs. Despite their hardware friendliness, STDP-based methods are difficult to scale to deep models required for real-world tasks. Accelerators (Liang et al., 2022; Frenkel & Indiveri, 2022) and (Park et al., 2019; Chen et al., 2024b; Guo et al., 2022; Wang et al., 2023; Lee et al., 2020b) explore approximate or local learning to mitigate BPTT overhead.

## B    DETAILS OF THE SUBSPACE-BASED ZEROTH-ORDER (SZO) TRAINING ALGORITHM

A detailed sketch of SZO is presented in Algorithm 1.

---

**Algorithm 1** Subspace-based zeroth-order (SZO) training

---

Initial weight $\theta_0$, loss $\mathcal{L}$, perturbation scale $\mu$, maximum iteration $T$, projection matrix $P = [v_1, v_2, ..., v_k]$, and dataset $\mathcal{D}$
**for** $t = 1, ..., T$ **do**
    Sample batch $\mathcal{B}_t \subset \mathcal{D}$
    Initialize the gradient $\hat{g}_t \leftarrow 0 \in \mathbb{R}^d$
    **if** use SZO-CGE **then**
        **for** $i = 1, ..., k$ **do**
            $\mathcal{L}^+ \leftarrow \mathcal{L}_{\mathcal{B}_t}(\theta_t + \mu v_i)$
            $\mathcal{L}^- \leftarrow \mathcal{L}_{\mathcal{B}_t}(\theta_t - \mu v_i)$
            $\hat{g}_t \leftarrow \hat{g}_t + \frac{\mathcal{L}^+ - \mathcal{L}^-}{2\mu} v_i$
        **end for**
    **else if** use SZO-RGE **then**
        **for** $i = 1, ..., q$ **do**
            Sample $z_i \sim \mathcal{N}(0, I_k)$
            $\mathcal{L}^+ \leftarrow \mathcal{L}_{\mathcal{B}_t}(\theta_t + \mu P z_i)$
            $\mathcal{L}^- \leftarrow \mathcal{L}_{\mathcal{B}_t}(\theta_t - \mu P z_i)$
            $\hat{g}_t \leftarrow \hat{g}_t + \frac{1}{q} \frac{\mathcal{L}^+ - \mathcal{L}^-}{2\mu} P z_i$
        **end for**
    **end if**
    Updated model $\theta_t$ through SGD with momentum $\theta_{t+1} = \theta_t - \eta \hat{g}_t + \beta(\theta_t - \theta_{t-1})$.
**end for**

---

## C    DETAILED RESULTS IN THE TRAINING FROM SCRATCH EXPERIMENT

Figure 7 and Figure 8 show the test accuracy curves of the BP-subspace and SZO-CGE/RGE methods in the training procedure on CIFAR-100 (Spike-ResNet18) and ImageNet1K (Spike-ResNet50), respectively.

## D    COMPARISON OF GRADIENT VARIANCE

We report the variance of gradient of all parameters estimated by full-space ZO (RGE, $q = 20$) on ANN and SNN with same architecture in Figure 9. It can be seen that ZO method indeed produces higher variance in SNN compared to ANN. This empirically validates the conclusion of Proposition 1 that the non-smooth nature of the Heaviside function is a significant source of gradient variance.

Figure 10 and Figure 11 show that the proposed subspace method reduces the variance of ZO gradient estimation for ANN (ReLU) and SNN, respectively. While the subspace strategy also lowers the ZO

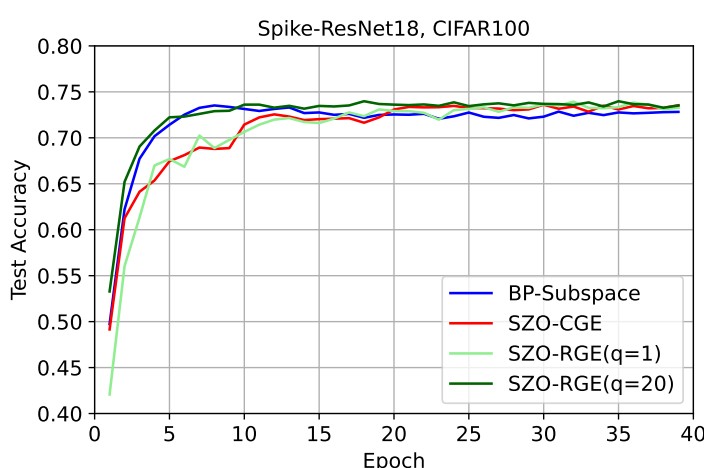

Figure 7: Test accuracy of Spike-ResNet18 on CIFAR100.

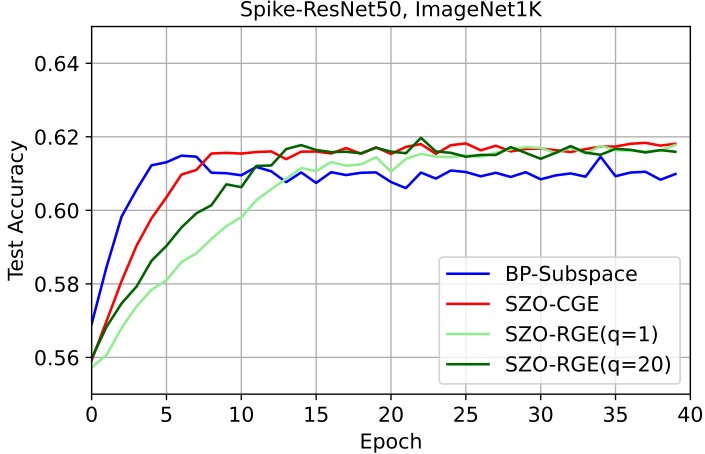

Figure 8: Test accuracy of Spike-ResNet50 on ImageNet1k.

variance in ANNs, it yields a much larger variance reduction in SNNs, which highlights its particular effectiveness in mitigating the Heaviside-induced instability.

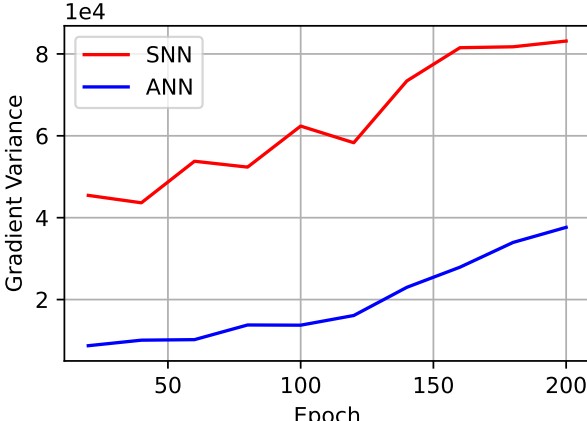

Figure 9: Gradient variance of full-space ZO on CIFAR10 with SNN (SpikeResNet20) and ANN (ResNet20).

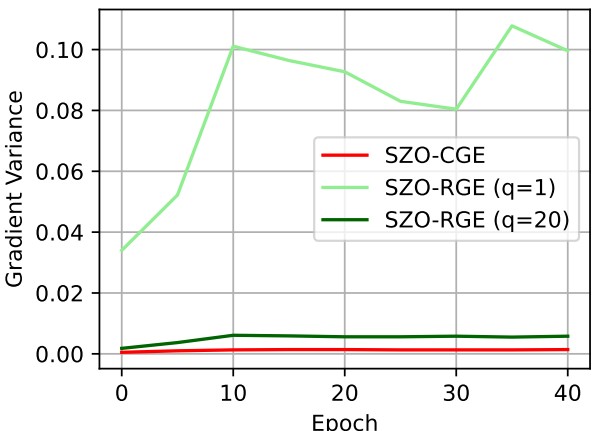

Figure 10: Gradient variance of diffenrent methods on CIFAR10 with ANN (ResNet20).

# E DETAILED RESULTS IN THE UNSUPERVISED MODEL ADAPTATION EXPERIMENT

We provide detailed results in the unsupervised model adaptation experiment in Table 5 to Table 10, in which the accuracy for each of the 15 corruptions are reported for the compared methods.

Table 5: Accuracy (%) on CIFAR10C dataset with corruption severity level 5 under *Reset* setting.

| Method | BP | Noise Gauss. | Shot | Impul. | Blur Defoc. Defoc. | Glass | Motion | Zoom | Weather Snow | Frost | Fog | Brit. | Digital Contr. | Elas. | Pix. | JPEG | Average Acc. |
|---|---|---|---|---|---|---|---|---|---|---|---|---|---|---|---|---|---|
| NoAdapt | X | 14.57 | 17.47 | 24.49 | 50.88 | 31.54 | 52.10 | 53.67 | 55.79 | 42.39 | 53.70 | 80.56 | 22.48 | 67.07 | 36.55 | 71.85 | 45.01 |
| BN | X | 53.79 | 54.13 | 57.78 | 73.37 | 56.85 | 71.00 | 74.20 | 62.45 | 61.89 | 68.67 | 79.24 | 61.87 | 70.01 | 66.22 | 71.43 | 65.53 |
| Tent | ✓ | 54.88 | 55.96 | 58.57 | 74.01 | 57.58 | 71.33 | 75.21 | 63.21 | 62.73 | 69.46 | 79.18 | 61.59 | 70.22 | 66.91 | 71.33 | 66.14 |
| BP-Subspace | ✓ | 67.45 | 67.24 | 64.58 | 74.15 | 58.31 | 72.41 | 75.45 | 70.23 | 69.75 | 70.52 | 82.35 | 24.34 | 71.27 | 71.78 | 75.45 | 67.69 |
| SZO-CGE | X | 70.97 | 73.94 | 65.70 | 75.19 | 44.49 | 68.32 | 75.47 | 70.10 | 71.08 | 70.04 | 83.50 | 19.33 | 70.03 | 72.65 | 75.52 | 67.09 |
| SZO-RGE(q=10) | X | 62.10 | 52.59 | 56.59 | 62.30 | 35.98 | 58.71 | 68.54 | 60.80 | 49.00 | 52.67 | 77.66 | 28.23 | 61.09 | 57.29 | 72.80 | 57.09 |

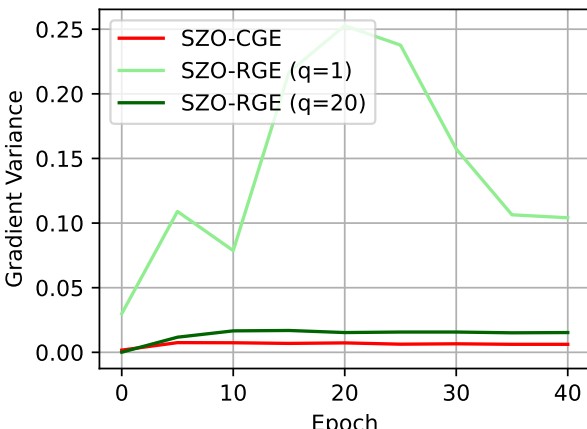

Figure 11: Gradient variance of diffenrent methods on CIFAR10 with SNN (SpikeResNet20).

Table 6: Accuracy (%) on CIFAR100C dataset with corruption severity level 5 under *Reset* setting.

| Method | BP | Noise Gauss. | Shot | Impul. | Blur Defoc. Defoc. | Glass | Motion | Zoom | Weather Snow | Frost | Fog | Brit. | Digital Contr. | Elas. | Pix. | JPEG | Average Acc. |
|---|---|---|---|---|---|---|---|---|---|---|---|---|---|---|---|---|---|
| NoAdapt | X | 8.01 | 9.55 | 9.71 | 29.85 | 21.51 | 38.16 | 36.05 | 41.89 | 28.72 | 32.53 | 59.36 | 10.39 | 51.91 | 22.56 | 46.91 | 29.81 |
| BN | X | 38.06 | 37.71 | 38.06 | 62.28 | 45.21 | 58.92 | 63.06 | 49.52 | 49.82 | 51.76 | 64.34 | 53.77 | 56.12 | 55.36 | 51.83 | 51.72 |
| Tent | ✓ | 39.22 | 39.36 | 39.71 | 62.92 | 45.76 | 59.42 | 63.77 | 50.44 | 50.09 | 52.22 | 65.14 | 53.85 | 56.60 | 56.00 | 52.21 | 52.45 |
| BP-Subspace | ✓ | 75.69 | 75.93 | 49.68 | 86.19 | 56.32 | 76.52 | 83.19 | 76.20 | 80.02 | 68.52 | 91.56 | 67.28 | 72.93 | 84.23 | 81.06 | 75.02 |
| SZO-CGE | X | 74.22 | 73.65 | 48.36 | 81.65 | 52.60 | 74.29 | 79.81 | 72.88 | 77.41 | 65.78 | 90.98 | 66.43 | 71.03 | 79.54 | 78.46 | 72.47 |
| SZO-RGE(q=10) | X | 57.79 | 59.13 | 47.89 | 70.32 | 52.00 | 67.21 | 71.02 | 62.27 | 63.48 | 59.83 | 75.45 | 61.37 | 63.98 | 68.62 | 65.11 | 63.03 |

Table 7: Accuracy (%) on ImageNetC dataset with corruption severity level 5 under *Reset* setting.

| Method | BP | Noise Gauss. | Shot | Impul. | Blur Defoc. Defoc. | Glass | Motion | Zoom | Weather Snow | Frost | Fog | Brit. | Digital Contr. | Elas. | Pix. | JPEG | Average Acc. |
|---|---|---|---|---|---|---|---|---|---|---|---|---|---|---|---|---|---|
| NoAdapt | X | 0.62 | 0.88 | 0.78 | 7.44 | 6.78 | 8.52 | 16.02 | 5.02 | 7.82 | 5.92 | 34.32 | 1.12 | 18.58 | 8.02 | 18.62 | 9.36 |
| BN | X | 5.82 | 7.40 | 6.02 | 11.80 | 13.10 | 17.54 | 28.38 | 16.42 | 19.92 | 28.40 | 48.20 | 9.16 | 39.58 | 36.16 | 30.10 | 21.20 |
| Tent | ✓ | 7.24 | 8.76 | 8.10 | 11.94 | 13.80 | 19.90 | 30.08 | 17.18 | 21.14 | 31.76 | 49.14 | 8.84 | 42.78 | 38.62 | 33.00 | 22.82 |
| BP-Subspace | ✓ | 45.64 | 37.64 | 33.00 | 14.98 | 18.65 | 23.12 | 33.45 | 19.36 | 25.98 | 30.00 | 50.26 | 5.85 | 45.77 | 46.18 | 46.53 | 31.76 |
| SZO-CGE | X | 50.55 | 40.80 | 35.56 | 15.62 | 19.65 | 22.87 | 33.80 | 19.69 | 27.15 | 29.67 | 50.28 | 6.60 | 46.45 | 46.98 | 48.37 | 32.94 |
| SZO-RGE(q=10) | X | 47.92 | 38.998 | 33.93 | 14.85 | 18.83 | 22.73 | 33.08 | 19.88 | 26.49 | 30.50 | 50.54 | 6.64 | 46.17 | 47.24 | 47.34 | 32.34 |

Table 8: Accuracy (%) on CIFAR10C dataset with corruption severity level 5 under *Continual* setting.

| Method | BP | Noise Gauss. | Shot | Impul. | Blur Defoc. Defoc. | Glass | Motion | Zoom | Weather Snow | Frost | Fog | Brit. | Digital Contr. | Elas. | Pix. | JPEG | Average Acc. |
|---|---|---|---|---|---|---|---|---|---|---|---|---|---|---|---|---|---|
| NoAdapt | X | 14.57 | 17.47 | 24.49 | 50.88 | 31.54 | 52.10 | 53.67 | 55.79 | 42.39 | 53.70 | 80.56 | 22.48 | 67.07 | 36.55 | 71.85 | 45.01 |
| BN | X | 53.79 | 54.13 | 57.78 | 73.37 | 56.85 | 71.00 | 74.20 | 62.45 | 61.89 | 68.67 | 79.24 | 53.77 | 70.01 | 66.22 | 71.43 | 65.53 |
| Tent | ✓ | 55.83 | 58.65 | 61.21 | 74.04 | 57.84 | 70.45 | 73.76 | 63.45 | 62.45 | 66.81 | 74.93 | 55.56 | 65.54 | 62.54 | 65.70 | 64.58 |
| BP-Subspace | ✓ | 70.48 | 75.57 | 69.91 | 78.40 | 63.50 | 75.59 | 78.57 | 75.54 | 76.32 | 73.43 | 85.67 | 45.31 | 66.60 | 76.32 | 79.69 | 72.73 |
| SZO-CGE | X | 74.09 | 79.31 | 70.74 | 80.14 | 64.68 | 76.17 | 80.52 | 76.10 | 77.06 | 75.44 | 86.43 | 66.60 | 75.84 | 78.02 | 80.29 | 76.10 |
| SZO-RGE(q=10) | X | 59.20 | 64.79 | 63.18 | 65.29 | 47.49 | 37.73 | 25.36 | 19.22 | 14.53 | 13.10 | 13.87 | 10.87 | 11.40 | 10.65 | 9.71 | 31.09 |

Table 9: Accuracy (%) on CIFAR100C dataset with corruption severity level 5 under *Continual* setting.

| Method | BP | Noise Gauss. | Shot | Impul. | Blur Defoc. Defoc. | Glass | Motion | Zoom | Weather Snow | Frost | Fog | Brit. | Digital Contr. | Elas. | Pix. | JPEG | Average Acc. |
|---|---|---|---|---|---|---|---|---|---|---|---|---|---|---|---|---|---|
| NoAdapt | X | 8.01 | 9.55 | 9.71 | 29.85 | 21.51 | 38.16 | 36.05 | 41.89 | 28.72 | 32.53 | 59.36 | 10.39 | 51.91 | 22.56 | 46.91 | 29.81 |
| BN | X | 38.06 | 37.71 | 38.06 | 62.28 | 45.21 | 58.92 | 63.06 | 49.52 | 49.82 | 51.76 | 64.34 | 53.77 | 56.12 | 55.36 | 51.83 | 51.72 |
| Tent | ✓ | 38.51 | 39.34 | 40.99 | 63.34 | 47.01 | 60.62 | 64.66 | 53.39 | 52.33 | 53.61 | 67.32 | 54.05 | 58.22 | 59.26 | 54.88 | 53.84 |
| BP-Subspace | ✓ | 55.37 | 69.93 | 59.26 | 84.31 | 65.30 | 80.21 | 87.03 | 80.47 | 84.13 | 76.04 | 94.13 | 81.23 | 80.60 | 87.75 | 86.52 | 78.15 |
| SZO-CGE | X | 62.56 | 79.04 | 63.56 | 88.84 | 65.79 | 81.85 | 86.51 | 80.61 | 83.81 | 75.93 | 94.05 | 80.10 | 79.39 | 86.66 | 86.59 | 79.69 |
| SZO-RGE(q=10) | X | 57.79 | 72.71 | 59.10 | 83.88 | 62.43 | 78.28 | 83.21 | 76.70 | 79.79 | 72.31 | 91.04 | 75.34 | 76.94 | 84.96 | 83.39 | 75.86 |

Table 10: Accuracy (%) on ImageNetC dataset with corruption severity level 5 under *Continual* setting.

| Method | BP | Noise Gauss. | Shot | Impul. | Blur Defoc. Defoc. | Glass | Motion | Zoom | Weather Snow | Frost | Fog | Brit. | Digital Contr. | Elas. | Pix. | JPEG | Average Acc. |
|---|---|---|---|---|---|---|---|---|---|---|---|---|---|---|---|---|---|
| NoAdapt | X | 0.62 | 0.88 | 0.78 | 7.44 | 6.78 | 8.52 | 16.02 | 5.02 | 7.82 | 5.92 | 34.32 | 1.12 | 18.58 | 8.02 | 18.62 | 9.36 |
| BN | X | 5.82 | 7.40 | 6.02 | 11.80 | 13.10 | 17.54 | 28.38 | 16.42 | 19.92 | 28.40 | 48.20 | 9.16 | 39.58 | 36.16 | 30.10 | 21.20 |
| Tent | ✓ | 7.58 | 10.12 | 8.76 | 8.38 | 6.06 | 5.54 | 4.84 | 1.38 | 0.86 | 0.72 | 1.50 | 0.38 | 0.96 | 0.68 | 0.60 | 3.89 |
| BP-Subspace | ✓ | 19.03 | 33.09 | 32.93 | 16.65 | 20.92 | 24.34 | 34.93 | 22.12 | 28.48 | 30.79 | 51.71 | 13.92 | 48.81 | 50.97 | 53.51 | 32.15 |
| SZO-CGE | X | 23.30 | 36.61 | 36.09 | 17.20 | 21.53 | 24.95 | 35.55 | 22.70 | 29.22 | 31.09 | 52.07 | 14.61 | 49.55 | 51.95 | 54.18 | 33.37 |
| SZO-RGE(q=10) | X | 21.76 | 35.46 | 34.98 | 17.07 | 21.32 | 25.03 | 35.59 | 23.04 | 29.05 | 31.19 | 51.91 | 15.07 | 49.38 | 52.13 | 54.36 | 33.16 |

## F  MORE ABLATION EXPERIMENTS

**Number of queries.**  We evaluated SZO-RGE under different numbers of queries, with perturbation scale fixed at $\mu = 1$. As shown in Table 11, SZO-RGE can achieve accuracy comparable to SZO-CGE even when $q$ is smaller than the subspace dimension $k = 60$.

Table 11: Accuracy (%) of SZO-RGE on training from scratch task with different number of queries.

| Dataset | Network | $q = 1$ | $q = 20$ | $q = 40$ | $q = 60$ |
|---------|---------|---------|----------|----------|----------|
| CIFAR10 | Spike-ResNet20 | 81.29±0.35 | 83.71±0.34 | 83.44±0.16 | 83.52±0.09 |
| CIFAR100 | Spike-ResNet18 | 73.77±0.26 | 73.79±0.22 | 73.90±0.24 | 73.87±0.17 |

We further evaluated the computational trade-off of SZO-RGE under different numbers of perturbations $q$ in the training from scratch task. We report the converged epoch, wall-clock converged time, and number of forward passes (#FP) required to reach the maximum accuracy. The results are shown in Table 12. While smaller $q$ yields faster training but lower accuracy, increasing $q$ improves the final accuracy at the cost of significantly longer training time and higher computational overhead.

**Comparison of one-point and two-point gradient estimation**  We have conducted additional experiments using single-point gradient estimation. The results in Table 13 show that one-point estimation can achieve comparable performance to two-point estimation on CIFAR100.

## G  MORE IMPLEMENTATION DETAILS

All network architectures utilize LIF neurons with $\tau = 1.1$ and $V_{th} = 1$. A triangle surrogate function is used in BPTT. Furthermore, we keep the running statistics of batch norm layers before each perturbation on training step $t$ the same.

In the training from scratch experiment, the model vectors used to construct the subspace are sampled from the BPTT training trajectory. During the BPTT training stage, the SGD optimizer is used for all datasets, with an initial learning rate of 0.1, momentum of 0.9, and a weight decay of 1e-4. The learning rate is decayed using a cosine method.

The SGD optimizer is also utilized in BP-Subspace and SZO for the training from scratch and continual training tasks, but the initial learning rate is set to 1, with momentum at 0.9 and no weight decay. In the SZO unsupervised adaptation experiment, a constant learning rate of 1 and 0.1 is used for *Reset* setting and *Continual* setting respectively.

The batch size is set to 128 for CIFAR10 and CIFAR100, 16 for DVS-Gesture, and 512 for ImageNet1K. For adaptation on corrupted datasets, the batch size is 100 for CIFAR10C and CIFAR100C, and 500 for ImageNetC. Common data augmentation methods, including random cropping and random horizontal flipping, are applied to static datasets. For DVS-Gesture, we integrate events and split each sample into 20 frames for the input of each time step.

## H  PROOF OF PROPOSITION 1

*Proof.*  First, we consider the Heaviside function $f_h(z) = \mathbb{I}(z > 0)$. Here, without loss of generality, we assume the firing threshold to be zero. From the symmetry and continuity assumption in Proposition 1, we have $\Pr(z > 0) = \Pr(z < 0) = \frac{1}{2}$. Hence the output of the Heaviside function is a Bernoulli distribution, i.e., $f_h(z) \sim \text{Bernoulli}(\frac{1}{2})$. Then, it follows that

$$\text{Var}\big(f_h(z)\big) = \frac{1}{4}. \tag{7}$$

Then, we consider the sigmoid function $f_s(z) = \frac{1}{1+e^{-z}}$. Using a first-order Taylor expansion around $z = 0$ we have $f_s(z) = \frac{1}{2} + \frac{1}{4}z + O(z^2)$. Then, we have

$$\text{Var}\big(f_s(z)\big) = \frac{1}{16}\text{Var}(z) + O(\sigma^4) = \frac{\sigma^2}{16} + o(\sigma^2). \tag{8}$$

Table 12: Comparison of computational overhead in the training-from-scratch task on CIFAR10 (SpikeResNet20) and CIFAR100 (SpikeResNet18).

| $q$ | CIFAR10 / SpikeResNet20 | | | | | CIFAR100 / SpikeResNet18 | | | | |
|---|---|---|---|---|---|---|---|---|---|---|
| | Converged epoch | Converged time (min) | #FP | #BP | Max Acc(%) | Converged epoch | Converged time (min) | #FP | #BP | Max Acc(%) |
| $q = 1$ | 39 | 22.8 | 45747 | 0 | 80.55 | 39 | 60.7 | 45747 | 0 | 73.47 |
| $q = 20$ | 33 | 224.3 | 529023 | 0 | 83.96 | 15 | 243.2 | 240465 | 0 | 73.55 |
| $q = 40$ | 24 | 331.0 | 760104 | 0 | 83.60 | 21 | 680.4 | 665091 | 0 | 73.64 |
| $q = 60$ | 29 | 621.0 | 1372019 | 0 | 83.61 | 15 | 708.6 | 709665 | 0 | 73.68 |

Table 13: Accuracy (%) comparison of one-point and two-point gradient estimation on training from scratch task.

| Dataset | Network | BP | BP-Subspace | SZO-CGE | SZO-CGE (one-point) |
|---|---|---|---|---|---|
| CIFAR10 | Spike-ResNet20 | 86.65 | 86.22 | 86.01 | 83.00 |
| CIFAR100 | Spike-ResNet18 | 74.42 | 73.37 | 73.42 | 73.28 |

Next, we consider the ReLU function $f_r(z) = \max\{0, z\}$. From the symmetry assumption in Proposition 1, we have

$$\mathbb{E}[f_r(z)] = \frac{1}{2}\mathbb{E}[|z|],$$

and

$$\mathbb{E}\left[f_r^2(z)\right] = \frac{1}{2}\mathbb{E}[z^2] = \frac{1}{2}\sigma^2.$$

Thus, it follows that

$$\mathrm{Var}\big(f_r(z)\big) = \frac{\sigma^2}{2} - \frac{1}{4}\Big(\mathbb{E}[|z|]\Big)^2. \tag{9}$$

Then, from the Cauchy–Schwarz inequality, we have $\mathbb{E}[|z|] \leq \sigma$. Hence, we have

$$\frac{\sigma^2}{4} \leq \mathrm{Var}(f_r(z)) \leq \frac{\sigma^2}{2}. \tag{10}$$

Finally, comparing across (7), (8) and (10) gives the final result. □

# I PROOF OF LEMMA 2

*Proof.* To simplify the discussion, we denote $w := P^\top\theta \in \mathbb{R}^k$, $l_x(w) := \mathcal{L}_x(Pw)$, $l_\mathcal{B}(w) := \mathcal{L}_\mathcal{B}(Pw)$ and $l(w) := \mathcal{L}(\theta)$, which means $g$ is the loss function in the subspace. So we will also have $\nabla l_x(w) = P^\top\nabla\mathcal{L}_x(Pw)$.

We first prove that $l_x(\theta)$ is a $L_P$-smooth function respect to $\theta$. From the definition of $l$ and the Assumption 1, for any $u, v \in \mathbb{R}^k$, we have the following inequality:

$$\|\nabla l_x(u) - \nabla l_x(v)\| = \|P^\top\left(\nabla\mathcal{L}_x(Pu) - \nabla\mathcal{L}_x(Pv)\right)\|$$
$$\leq \|P\|_{\mathrm{op}} \cdot L\|P(u - v)\| \leq L\|P\|_{\mathrm{op}}^2\|u - v\|,$$

where $\|\cdot\|_{\mathrm{op}}$ is the operator spectral norm. So $l_x$ is a $L_P$-smooth function with $L_p := L\|P\|_{\mathrm{op}}^2$.

Then we give the estimation error between $\hat{\nabla}l_\mathcal{B}(w)$ and $P^\top\nabla\mathcal{L}_\mathcal{B}(Pw)$. By Mean Value Theorem, we have,

$$l_x(w + \mu e_i) = l_x(w) + \mu\partial_i l_x(w) + \frac{\mu^2}{2}\partial_i^2 l_x(\xi_1),$$

$$l_x(w - \mu e_i) = l_x(w) - \mu\partial_i l_x(w) + \frac{\mu^2}{2}\partial_i^2 l_x(\xi_2),$$

$$\Rightarrow \hat{\nabla}_i l_x(w) = \frac{l_x(w + \mu e_i) - l_x(w - \mu e_i)}{2\mu} = \partial_i l_x(w) + \frac{\mu}{4}\left(\partial_i^2 l_x(\xi_1) - \partial_i^2 l_x(\xi_2)\right),$$

where $e_i$ is the standard basis in $\mathbb{R}^k$, $\xi_1 = \lambda_1 w + (1 - \lambda_1)(w + \mu e_i)$, $\xi_2 = \lambda_2 w + (1 - \lambda_2)(w - \mu e_i)$ and $\lambda_1, \lambda_2 \in (0, 1)$. Because $g_x$ is $L_P$-smooth function, and the operation norm of the Hessian

matrix $\nabla^2 l_x(\xi)$ is less than $L_P$, which means $|\partial_i^2 l_x(\xi)| \leq \|\nabla^2 l_x(\xi)\|_{\text{op}} \leq L$ for $\xi_1, \xi_2$. So we have

$$|\hat{\nabla}_i l_x(w) - \partial_i l_x(w)| \leq \frac{\mu}{4}(|\partial_i^2 l_x(\xi_1)| + |\partial_i^2 l_x(\xi_2)|) \leq \frac{\mu L_P}{2},$$

for each standard basis $e_i$. From the definition $l_{\mathcal{B}}(w) = \frac{1}{B}\sum_{x_i \in \mathcal{B}} l_{x_i}(w)$ and Jensen's inequality, we have

$$|\hat{\nabla}_i l_{\mathcal{B}}(w) - \partial_i l_{\mathcal{B}}(w)| \leq \frac{1}{B}\sum_{x_i \in \mathcal{B}} |\hat{\nabla}_i l_{x_i}(w) - \partial_i l_{x_i}(w)| \leq \frac{\mu L_P}{2}.$$

From the definition, we also have $\partial_i l_x(w) = P_i^\top \nabla \mathcal{L}_x(Pw)$ and $\nabla l_x(w) = P^\top \nabla \mathcal{L}_x(Pw)$, where $P_i$ is the $i$-th column of the projection matrix $P$. Then we have

$$\|\hat{\nabla} l_{\mathcal{B}}(w) - P^\top \nabla \mathcal{L}_{\mathcal{B}}(Pw)\|^2 = \sum_{i=1}^{k} \left(\hat{\nabla}_i l_{\mathcal{B}}(w) - P_i^\top \nabla \mathcal{L}_{\mathcal{B}}(Pw)\right)^2 \leq \frac{k\mu^2 L_P^2}{4}.$$

Then from the above calculate and the assumption, for a given $w_t$ and $\theta_t = Pw_t$ and a mini-batch $\mathcal{B}$, by Cauchy-Schwarz inequality, we have

$$\begin{aligned}
&\|\hat{\nabla}^{\text{SZO}}\mathcal{L}_{\mathcal{B}}(\theta_t) - \nabla \mathcal{L}_{\mathcal{B}}(\theta_t)\|^2 \\
=&\|P\hat{\nabla} l_{\mathcal{B}}(w_t) - \nabla \mathcal{L}_{\mathcal{B}}(\theta_t)\|^2 \\
=&\|P\hat{\nabla} l_{\mathcal{B}}(w_t) - PP^\top \nabla \mathcal{L}_{\mathcal{B}}(Pw_t) + PP^\top \nabla \mathcal{L}_{\mathcal{B}}(Pw_t) - \nabla \mathcal{L}_{\mathcal{B}}(\theta_t)\|^2 \\
\leq&2\|P\hat{\nabla} l_{\mathcal{B}}(w_t) - PP^\top \nabla \mathcal{L}_{\mathcal{B}}(Pw_t)\|^2 + 2\|PP^\top \nabla \mathcal{L}_{\mathcal{B}}(Pw_t) - \nabla \mathcal{L}_{\mathcal{B}}(\theta_t)\|^2 \\
\leq&\|P\|_{\text{op}}^2 k\mu^2 L_P^2/2 + 2\varepsilon^2 = k\mu^2 L^2\|P\|_{\text{op}}^4/2 + 2\varepsilon^2.
\end{aligned}$$

The proof is completed. $\qquad\square$

## J   PROOF OF THEOREM 3

To analyze the convergence rate in the presence of momentum, we quantify the one-step difference between the gradient evaluated at the parameter without momentum, $\theta_t$, and the gradient evaluated at the momentum-adjusted parameter, $m_t$.

**Lemma 5** (One-step momentum difference). *Suppose the loss function $\mathcal{L}$ and the projection matrix $P$ satisfy Assumption 1, and let $\mathcal{B}$ be a mini-batch of size $B$. Under the gradient estimator and parameter update rule given in equations (5) and (6), the following bound holds:*

$$\|\nabla \mathcal{L}(m_t) - \nabla \mathcal{L}(\theta_t)\|^2 \leq \frac{L^2\beta^2\eta^2\left(2G^2 + 2\sigma^2/B + k\mu^2 L^2\|P\|_{op}^4 + 4\varepsilon^2\right)}{(1-\beta)^4},$$

*where $\|\cdot\|_{op}$ denotes the operator norm.*

*Proof of Lemma 5.* Because the loss function $\mathcal{L}_x(\theta)$ is $L$-smooth respective to $\theta$, so $\mathcal{L}(\theta)$ is also $L$-smooth respective to $\theta$, we have

$$\|\nabla \mathcal{L}(m_t) - \nabla \mathcal{L}(\theta_t)\|^2 \leq L^2\|m_t - \theta_t\| = L^2\|p_k\|^2 = \frac{L^2\beta^2}{(1-\beta)^2}\|\theta_t - \theta_{t-1}\|^2$$

Then from the definition of $\theta_t - \theta_{t-1}$, we have

$$\begin{aligned}
\theta_t - \theta_{t-1} &= \beta(\theta_{t-1} - \theta_{t-2}) - \eta\hat{\nabla}^{\text{SZO}}\mathcal{L}_{\mathcal{B}}(\theta_{t-1}) \\
&= -\eta\sum_{\tau=0}^{t-1} \beta^{t-1-\tau}\hat{\nabla}^{\text{SZO}}\mathcal{L}_{\mathcal{B}}(\theta_\tau) = -\eta\sum_{\tau=0}^{t-1} \beta^\tau\hat{\nabla}^{\text{SZO}}\mathcal{L}_{\mathcal{B}}(\theta_{t-1-\tau}).
\end{aligned}$$

We define $\Gamma_{t-1} = \sum_{\tau=0}^{t-1} \beta^\tau = \frac{1-\beta^t}{1-\beta}$, so we know $\sum_{\tau=0}^{t-1} \frac{\beta^\tau}{\Gamma_{t-1}} = 1$ and we can apply Jensen's inequailty to obtain

$$
\|\theta_t - \theta_{t-1}\|^2 = \|\eta \sum_{\tau=0}^{t-1} \frac{\beta^\tau}{\Gamma_{t-1}} \hat{\nabla}^{\text{SZO}} \mathcal{L}_{\mathcal{B}}(\theta_{t-1-\tau})\|^2 \Gamma_{t-1}^2
$$

$$
\leq \eta^2 \Gamma_{t-1}^2 \sum_{\tau=0}^{t-1} \frac{\beta^\tau}{\Gamma_{t-1}} \|\hat{\nabla}^{\text{SZO}} \mathcal{L}_{\mathcal{B}}(\theta_{t-1-\tau})\|^2
$$

$$
= \eta^2 \Gamma_{t-1} \sum_{\tau=0}^{t-1} \beta^\tau \|\hat{\nabla}^{\text{SZO}} \mathcal{L}_{\mathcal{B}}(\theta_{t-1-\tau})\|^2
$$

$$
\leq \eta^2 \Gamma_{t-1} \sum_{\tau=0}^{t-1} \beta^\tau \|\hat{\nabla}^{\text{SZO}} \mathcal{L}_{\mathcal{B}}(\theta_{t-1-\tau})\|^2
$$

Besides, from the result given in Lemma 2 and by Cauchy-Schwarz inequality, we have

$$
\|\hat{\nabla}^{\text{SZO}} \mathcal{L}_{\mathcal{B}_t}(\theta_t)\|^2 \leq 2\|\hat{\nabla}^{\text{SZO}} \mathcal{L}_{\mathcal{B}_t}(\theta_t) - \mathcal{L}_{\mathcal{B}_t}(\theta_t)\|^2 + 2\|\nabla \mathcal{L}_{\mathcal{B}_t}(\theta_t)\|^2
$$
$$
\leq 2\|\nabla \mathcal{L}_{\mathcal{B}_t}(\theta_t)\|^2 + k\mu^2 L^2 \|P\|_{\text{op}}^4 + 4\varepsilon^2. \tag{11}
$$

Combining with the Assumption 1, we have

$$
\mathbb{E}_{\mathcal{B}_t}[\|\theta_t - \theta_{t-1}\|^2] \leq \eta^2 \Gamma_{t-1} \sum_{\tau=0}^{t-1} \beta^\tau \left( 2\mathbb{E}_{\mathcal{B}_t}[\|\nabla \mathcal{L}_{\mathcal{B}_t}(\theta_t)\|^2] + k\mu^2 L^2 \|P\|_{\text{op}}^4 + 4\varepsilon^2 \right)
$$

$$
\leq \eta^2 \Gamma_{t-1}^2 \left( 2(G^2 + \sigma^2/B) + k\mu^2 L^2 \|P\|_{\text{op}}^4 + 4\varepsilon^2 \right)
$$

$$
\leq \frac{\eta^2 \left( 2G^2 + 2\sigma^2/B + k\mu^2 L^2 \|P\|_{\text{op}}^4 + 4\varepsilon^2 \right)}{(1-\beta)^2}.
$$

Then we have

$$
\|\nabla \mathcal{L}(m_t) - \nabla \mathcal{L}(\theta_t)\|^2 = \mathbb{E}_{\mathcal{B}_t}[\|\nabla \mathcal{L}(m_t) - \nabla \mathcal{L}(\theta_t)\|^2] \leq \frac{L^2 \beta^2}{(1-\beta)^2} \mathbb{E}_{\mathcal{B}_t}[\|\theta_t - \theta_{t-1}\|^2]
$$

$$
\leq \frac{L^2 \beta^2 \eta^2 \left( 2G^2 + 2\sigma^2/B + k\mu^2 L^2 \|P\|_{\text{op}}^4 + 4\varepsilon^2 \right)}{(1-\beta)^4}.
$$

This complete the proof. $\qquad\square$

Combining the result from Lemma 2 and 5, we can complete the proof of Theorem 3.

*Proof of Theorem 3.* Because the loss function $\mathcal{L}_x(\theta)$ is $L$-smooth respective to $\theta$, so $\mathcal{L}(\theta)$ is also $L$-smooth respective to $\theta$. Combining $L$-smooth property of $\mathcal{L}(\theta)$ with the parameter update equation (6) under a mini-batch $\mathcal{B}_t$ and take the expectation on $\mathcal{B}_t$, we have

$$
\mathbb{E}_{\mathcal{B}_t}[\mathcal{L}(m_{t+1})] \leq \mathcal{L}(m_t) - \frac{\eta}{1-\beta} \nabla \mathcal{L}(m_t)^\top \mathbb{E}_{\mathcal{B}_t}[\hat{\nabla}^{\text{SZO}} \mathcal{L}_{\mathcal{B}_t}(\theta_t)] + \frac{L}{2} \mathbb{E}_{\mathcal{B}_t}[\|\frac{\eta}{1-\beta} \hat{\nabla}^{\text{SZO}} \mathcal{L}_{\mathcal{B}_t}(\theta_t)\|^2]. \tag{12}
$$

For the second term in the RHS, we have

$$
\nabla \mathcal{L}(m_t)^\top \left( -\frac{\eta}{1-\beta} \mathbb{E}_{\mathcal{B}_t}[\hat{\nabla}^{\text{SZO}} \mathcal{L}_{\mathcal{B}_t}(\theta_t)] \right)
$$

$$
= \left( \nabla \mathcal{L}(m_t) - \mathbb{E}_{\mathcal{B}_t}[\hat{\nabla}^{\text{SZO}} \mathcal{L}_{\mathcal{B}_t}(\theta_t)] \right)^\top \left( -\frac{\eta}{1-\beta} \mathbb{E}_{\mathcal{B}_t}[\hat{\nabla}^{\text{SZO}} \mathcal{L}_{\mathcal{B}_t}(\theta_t)] \right)
$$

$$
- \frac{\eta}{1-\beta} \|\mathbb{E}_{\mathcal{B}_t}[\hat{\nabla}^{\text{SZO}} \mathcal{L}_{\mathcal{B}_t}(\theta_t)]\|^2
$$

$$
\leq \frac{1}{2L} \|\nabla \mathcal{L}(m_t) - \mathbb{E}_{\mathcal{B}_t}[\hat{\nabla}^{\text{SZO}} \mathcal{L}_{\mathcal{B}_t}(\theta_t)]\|^2 + \left( \frac{L\eta^2}{2(1-\beta)^2} - \frac{\eta}{1-\beta} \right) \|\mathbb{E}_{\mathcal{B}_t}[\hat{\nabla}^{\text{SZO}} \mathcal{L}_{\mathcal{B}_t}(\theta_t)]\|^2 \tag{13}
$$

$$
\leq \frac{1}{L} \|\nabla \mathcal{L}(m_t) - \nabla \mathcal{L}(\theta_t)\|^2 + \frac{1}{L} \|\nabla \mathcal{L}(\theta_t) - \mathbb{E}_{\mathcal{B}_t}[\hat{\nabla}^{\text{SZO}} \mathcal{L}_{\mathcal{B}_t}(\theta_t)]\|^2
$$

$$
+ \left( \frac{L\eta^2}{2(1-\beta)^2} - \frac{\eta}{1-\beta} \right) \|\mathbb{E}_{\mathcal{B}_t}[\hat{\nabla}^{\text{SZO}} \mathcal{L}_{\mathcal{B}_t}(\theta_t)]\|^2.
$$

Besides, from the result given in Lemma 2 and by Cauchy-Schwarz inequality, we have

$$\|\hat{\nabla}^{\text{SZO}}\mathcal{L}_{\mathcal{B}_t}(\theta_t)\|^2 \leq 2\|\hat{\nabla}^{\text{SZO}}\mathcal{L}_{\mathcal{B}_t}(\theta_t) - \mathcal{L}_{\mathcal{B}_t}(\theta_t)\|^2 + 2\|\nabla\mathcal{L}_{\mathcal{B}_t}(\theta_t)\|^2$$
$$\leq 2\|\nabla\mathcal{L}_{\mathcal{B}_t}(\theta_t)\|^2 + k\mu^2 L^2\|P\|_{\text{op}}^4 + 4\varepsilon^2. \tag{14}$$

And we also have

$$\|\mathbb{E}_{\mathcal{B}_t}[\hat{\nabla}^{\text{SZO}}\mathcal{L}_{\mathcal{B}_t}(\theta_t)]\|^2 \leq 2\|\mathbb{E}_{\mathcal{B}_t}[\hat{\nabla}^{\text{SZO}}\mathcal{L}_{\mathcal{B}_t}(\theta_t)] - \nabla\mathcal{L}(\theta_t)\|^2 + 2\|\nabla\mathcal{L}(\theta_t)\|^2$$
$$\leq 2\mathbb{E}_{\mathcal{B}_t}[\|\hat{\nabla}^{\text{SZO}}\mathcal{L}_{\mathcal{B}_t}(\theta_t) - \nabla\mathcal{L}_{\mathcal{B}_t}(\theta_t)\|^2] + 2\|\nabla\mathcal{L}(\theta_t)\|^2 \tag{15}$$
$$\leq 2\|\nabla\mathcal{L}(\theta_t)\|^2 + k\mu^2 L^2\|P\|_{\text{op}}^4 + 4\varepsilon^2.$$

Combining (12), (13), (14), (15) and result in Lemma 5, we have

$$\mathbb{E}_{\mathcal{B}_t}[\mathcal{L}(m_{t+1})] - \mathcal{L}(m_t)$$
$$\leq \frac{1}{L}\|\nabla\mathcal{L}(m_t) - \nabla\mathcal{L}(\theta_t)]\|^2 + \frac{1}{L}\mathbb{E}_{\mathcal{B}_t}[\|\nabla\mathcal{L}_{\mathcal{B}_t}(\theta_t) - \hat{\nabla}^{\text{SZO}}\mathcal{L}_{\mathcal{B}_t}(\theta_t)\|^2]$$
$$\quad + \left(\frac{L\eta^2}{2(1-\beta)^2} - \frac{\eta}{1-\beta}\right)\mathbb{E}_{\mathcal{B}_t}[\|\hat{\nabla}^{\text{SZO}}\mathcal{L}_{\mathcal{B}_t}(\theta_t)\|^2] + \frac{L\eta^2}{2(1-\beta)^2}\mathbb{E}_{\mathcal{B}_t}[\|\hat{\nabla}^{\text{SZO}}\mathcal{L}_{\mathcal{B}_t}(\theta_t)\|^2]$$
$$\leq \frac{L\beta^2\eta^2\left(2G^2 + 2\sigma^2/B + k\mu^2 L^2\|P\|_{\text{op}}^4 + 4\varepsilon^2\right)}{(1-\beta)^4} + \frac{k\mu^2 L\|P\|_{\text{op}}^4}{2} + \frac{2\varepsilon^2}{L}$$
$$\quad + \left(\frac{L\eta^2}{2(1-\beta)^2} - \frac{\eta}{1-\beta}\right)\left(2\|\nabla\mathcal{L}(\theta_t)\|^2 + k\mu^2 L^2\|P\|_{\text{op}}^4 + 4\varepsilon^2\right)$$
$$\quad + \frac{L\eta^2}{2(1-\beta)^2}\left(2\|\nabla\mathcal{L}(\theta_t)\|^2 + 2\sigma^2/B + k\mu^2 L^2\|P\|_{\text{op}}^4 + 4\varepsilon^2\right)$$
$$\leq \frac{L\beta^2\eta^2\left(2G^2 + 2\sigma^2/B + k\mu^2 L^2\|P\|_{\text{op}}^4 + 4\varepsilon^2\right)}{(1-\beta)^4} + \frac{k\mu^2 L\|P\|_{\text{op}}^4}{2} + \frac{2\varepsilon^2}{L}$$
$$\quad + \left(\frac{L\eta^2}{(1-\beta)^2} - \frac{\eta}{1-\beta}\right)\left(2\|\nabla\mathcal{L}(\theta_t)\|^2 + k\mu^2 L^2\|P\|_{\text{op}}^4 + 4\varepsilon^2\right) + \frac{L\eta^2}{B(1-\beta)^2}\sigma^2$$

Re-arrange the above inequality, we have

$$2\left(\frac{\eta}{1-\beta} - \frac{L\eta^2}{(1-\beta)^2}\right)\|\nabla\mathcal{L}(\theta_t)\|^2$$
$$\leq \mathcal{L}(m_t) - \mathbb{E}_{\mathcal{B}_t}[\mathcal{L}(m_{t+1})] + \frac{2L\beta^2\eta^2}{(1-\beta)^4}G^2 + \left(\frac{4L\beta^2\eta^2}{(1-\beta)^4} + \frac{1}{L} + \frac{4L\eta^2}{(1-\beta)^2} - \frac{4\eta}{1-\beta}\right)\varepsilon^2$$
$$\quad + \left(\frac{L\beta^2\eta^2}{(1-\beta)^4} + \frac{1}{2L} + \frac{L\eta^2}{(1-\beta)^2} - \frac{\eta}{1-\beta}\right)k\mu^2 L^2\|P\|_{\text{op}}^4 + \left(\frac{2L\beta^2\eta^2}{B(1-\beta)^4} + \frac{2L\eta^2}{B(1-\beta)^2}\right)\sigma^2.$$

Sum the above inequality from $t = 0$ to $T - 1$ with iterative expectation with $\mathcal{B}_0, ..., \mathcal{B}_{t-1}$, and recall $0 < \varepsilon \leq \eta$, $0 < \mu \leq \eta$ from the assumption, we have

$$\sum_{t=0}^{T-1} 2\left(\frac{\eta}{1-\beta} - \frac{L\eta^2}{(1-\beta)^2}\right)\mathbb{E}_{\mathcal{B}_0,...,\mathcal{B}_{t-1}}[\|\nabla\mathcal{L}(\theta_t)\|^2]$$
$$\leq \sum_{t=0}^{T-1}\left[\mathbb{E}_{\mathcal{B}_0,...,\mathcal{B}_{t-1}}[\mathcal{L}(m_t)] - \mathbb{E}_{\mathcal{B}_0,...,\mathcal{B}_t}[\mathcal{L}(m_{t+1})]\right] + \sum_{t=0}^{T-1}\frac{2L\beta^2\eta^2}{(1-\beta)^4}G^2$$
$$\quad + \sum_{t=0}^{T-1}\left(\frac{4L\beta^2\eta^4}{(1-\beta)^4} + \frac{\eta^2}{L} + \frac{4L\eta^4}{(1-\beta)^2} - \frac{4\eta^3}{1-\beta}\right) + \sum_{t=0}^{T-1}\left(\frac{2L\beta^2\eta^2}{B(1-\beta)^4} + \frac{2L\eta^2}{B(1-\beta)^2}\right)\sigma^2$$
$$\quad + \sum_{t=0}^{T-1}\left(\frac{L\beta^2\eta^4}{(1-\beta)^4} + \frac{\eta^2}{2L} + \frac{L\eta^4}{(1-\beta)^2} - \frac{\eta^3}{1-\beta}\right)kL^2\|P\|_{\text{op}}^4$$

Using $\mathcal{L}(\theta^*)$ to replace $\mathbb{E}_{\mathcal{B}_0,\ldots,\mathcal{B}_{T-1}}[\mathcal{L}(m_T)]$ because of $\mathcal{L}(\theta^*) \leq \mathbb{E}_{\mathcal{B}_0,\ldots,\mathcal{B}_{T-1}}[\mathcal{L}(m_T)]$, and $m_0 = \theta_0$, then divided both sides with $\sum_{t=0}^{T-1} 2\left(\frac{\eta}{1-\beta} - \frac{L\eta^2}{(1-\beta)^2}\right) = \frac{2(\sqrt{T}-1)}{L}$, we have

$$\frac{1}{T}\sum_{t=0}^{T-1}\mathbb{E}_{\mathcal{B}_0,\ldots,\mathcal{B}_{t-1}}\left[\|\nabla\mathcal{L}(\theta_t)\|^2\right] \leq \frac{L[\mathcal{L}(\theta_0) - \mathcal{L}(\theta^*)]}{2(\sqrt{T}-1)} + \frac{\beta^2 G^2}{(1-\beta)^2(\sqrt{T}-1)} + \frac{(\beta^2 + (1-\beta)^2)\sigma^2}{(1-\beta)^2(\sqrt{T}-1)B}$$

$$+ \left(\frac{2\beta^2 + 2(1-\beta)^2}{L^2 T(\sqrt{T}-1)} + \frac{(1-\beta)^2}{2L^2(\sqrt{T}-1)} + \frac{2(1-\beta)^2}{L^2(T-\sqrt{T})}\right)$$

$$+ \left(\frac{\beta^2 + (1-\beta)^2}{2T(\sqrt{T}-1)} + \frac{(1-\beta)^2}{4(\sqrt{T}-1)} + \frac{(1-\beta)^2}{2(T-\sqrt{T})}\right)k\|P\|_{\text{op}}^4.$$

The convergence rate can be represented as the leading term, which is $\mathcal{O}\left(\frac{k(1-\beta)^2}{\sqrt{T}} + \left(\frac{\beta}{1-\beta}\right)^2\frac{\sigma^2}{B\sqrt{T}}\right)$.

The proof is complete. $\square$

## K  PROOF OF THEOREM 4

*Proof.* From the definition of $\hat{\nabla}^{\text{SZO}}\mathcal{L}_{\mathcal{B}}(\theta)$, we have that when $\mu \to 0$, the expectation and covariance of $\hat{\nabla}^{\text{SZO}}\mathcal{L}_{\mathcal{B}}(\theta)$ can be expressed as

$$\mathbb{E}[\lim_{\mu\to 0}\hat{\nabla}^{\text{SZO}}\mathcal{L}_{\mathcal{B}}(\theta)] = PP^T\nabla\mathcal{L}(\theta) = \bar{P}\nabla\mathcal{L}(\theta), \text{ and } \text{Cov}[\lim_{\mu\to 0}\hat{\nabla}^{\text{SZO}}\mathcal{L}_{\mathcal{B}}(\theta)] = \bar{P}\Sigma\bar{P}^\top.$$

The mini-batch SGD update rule for the SZO with subspace is defined as

$$\theta_{t+1} = \theta_t - \eta\hat{\nabla}^{\text{SZO}}\mathcal{L}(\theta). \tag{16}$$

When $\eta \to 0$, the discrete dynamics of (16) can be approximated by the following SDE (Oksendal, 2013; Stephan et al., 2017)

$$d\theta(t) = -\bar{P}\nabla\mathcal{L}(\theta)dt + \sqrt{\eta}\bar{P}\Sigma^{1/2}dW(t), \tag{17}$$

with drift $\bar{P}\nabla\mathcal{L}(\theta)$ and diffusion covariance $\bar{P}\Sigma\bar{P}$. With the use of the quadratic Lyapunov function $V(\theta) = \frac{1}{2}\|\theta - \theta^*\|^2$, for the SDE (17) the infinitesimal generator is

$$L_V = -(\nabla\mathcal{L}(\theta))^\top\bar{P}(\theta - \theta^*) + \frac{\eta}{2}\text{Tr}(\bar{P}\Sigma\bar{P}^\top).$$

Then following the Lyapunov-based stability theory, we can derive the sufficient condition for stability of SZO as

$$(\nabla\mathcal{L}(\theta))^\top\bar{P}(\theta - \theta^*) > \frac{\eta}{2}\text{Tr}(\bar{P}\Sigma_{\text{ZO}}\bar{P}^T).$$

Using the similar process, we can derive the the sufficient condition for stability of CGE as

$$(\nabla\mathcal{L}(\theta))^\top(\theta - \theta^*) > \frac{\eta}{2}\text{Tr}(\Sigma).$$

Furthermore, under the assumption of local strong monotonicity in $\mathcal{V}(\theta^*)$ with constant $\rho$, for the SZO method, we have:

$$(\nabla\mathcal{L}(\theta))^T\bar{P}(\theta - \theta^*) \geq \rho\|\bar{P}(\theta - \theta^*)\|^2.$$

Then we define $r := \sup_{\theta\in\mathcal{V}(\theta^*)}\|\bar{P}(\theta - \theta^*)\|$, and we have

$$L_V \leq -\rho\|\bar{P}(\theta - \theta^*)\|^2 + \frac{\eta k\sigma^2}{2B} \leq -\rho r^2 + \frac{\eta k\sigma^2}{2B}. \tag{18}$$

Then, a sufficient condition for local mean-square stability, i.e., $L_V \leq 0$, is given by

$$\eta < \frac{2B\rho r^2}{k\sigma^2}.$$

In comparison, for the CGE method, a similar condition for mean-square stability can be derived as

$$\eta < \frac{2B\rho r^2}{d\sigma^2}. \tag{19}$$

Thus, subspace-based ZO training can largely relax the condition for local stability when $k \ll d$. For example, the step-size for stability can be relaxed from $\eta < \mathcal{O}(1/d)$ to $\beta < \mathcal{O}(1/k)$. $\square$

