# OpenReview forum: "Zeroth-Order Forward-Only SNN Training Inspiring Neuromorphic On-Chip Learning"
_ICLR.cc/2026/Conference — Submitted to ICLR 2026_

### Official Review · Reviewer_1k7D · 2025-10-26

**Soundness:** 2
**Presentation:** 1
**Contribution:** 2
**Rating:** 2
**Confidence:** 5

**Summary:**

This papaer propose an parameter-effcient learning zeroth order learning method on SNN, possibly aiming at edge case learning.

**Strengths:**

1. Clear motivation. This work studies zeroth-order learning in SNN by first analyzing the variance introduced by the Heaviside function, then proposes multi-sampling and subspace-based learning to reduce it, two typical methods in zeroth-order learning.
2. Clear problem formulation and notations.
3. The author provides a dedicated analysis of the efficiency and superiority of the proposed zeroth-order learning method.

**Weaknesses:**

1. The paper lacks adequate discussion of existing methods that combine subspace-based fine-tuning with zeroth-order learning in ANNs [1-4]. The related work section in the appendix appears to **intentionally separate subspace-based learning from BP-free methods.** There is only one line mentioning ZO+subspace learning, stating "it has been recently shown that ZO optimization can effectively finetune LLMs," without explicitly describing how these methods fine-tune LLMs (through subspace learning techniques). There is a section for subspace method but also not connected to how these method is adopted by ZO-based methods. The authors should provide a detailed comparison with these prior works. Currently, the primary distinction appears to be the choice of PCA over LoRA or stochastic projection matrices used in prior ANN work. Especially given the  concerns regarding theoretical contribution (see W.2), a more thorough comparative analysis is needed.
2. **Assumption 1, Point 1** is problematic and undermines the theoretical foundation. The authors assume that $\mathcal{L}_x(\theta)$ is $L$-smooth with respect to $\theta$, yet the forward pass of SNNs is inherently discrete, even when employing surrogate gradients. This fundamental tension is not acknowledged as an approximation choice or limitation. *While assuming something that **might not** exist is acceptable in theoretical work, assuming something that **demonstrably does not** exist without any justification or discussion is problematic.* Consequently, the theoretical analysis effectively reduces to proving that combining subspace methods with zeroth-order optimization on *ANNs* (where the loss function may genuinely be $L$-smooth) can reduce variance and improve convergence rates—a result extensively studied in the ANN literature [1-4]. These prior works employ similar analytical frameworks assuming $L$-smoothness, exploitable local structure, and bounded gradient variance. I acknowledge that **Proposition 1** genuinely addresses SNNs by characterizing the increased variance when transitioning from continuous activation functions to the Heaviside function. However, this result does not lead to any subsequent analysis that meaningfully respects or leverages the discrete nature of SNNs. The remainder of the paper relies on assumptions appropriate for the ANN case, limiting the theoretical contribution.
3. Limited practical validation
    - *Missing Edge Device Evaluation:* Despite targeting edge device training, the paper provides no actual edge device performance evaluation, making it difficult to assess the real-world potential of the proposed algorithm.
    - *Unrealistic Operating Regimes and Missing Baselines:* In practice, edge deployments may only operate at time steps and batch sizes corresponding to the leftmost region of Figure 5's x-axis. Furthermore, the paper lacks comparison with gradient checkpointing methods that achieve $\mathcal{O}(\sqrt{n})$ memory scaling (where $n$ denotes layers or timesteps) with $\mathcal{O}(n)$ extra recomputation—typically a 1.5× slowdown that significantly reduces memory consumption. Given that SZO-CGE is slower than BP (and possibly comparable to SZO-RGE with larger $q$), especially in training-from-scratch settings, BP with checkpointing (possibly combined with parameter-efficient learning) appears to be a competitive alternative that should be evaluated.
    - *Mixed-Precision Training Not Addressed:* Edge training typically uses mixed-precision to further reduce memory, but this is not clearly discussed in the main paper. In such settings, parameters and their buffers (first and second-order momentum) are often maintained in full-precision for stability. While ZO methods save activation memory, parameters still occupy consistent memory throughout training. Therefore, the memory reduction may not be as significant as Figure 5 suggests, and this should be addressed.
4. Lack of justification for subspace selection. The authors choose PCA for parameter-efficient learning with theoretical analysis but provide no empirical justification for this choice. For continued learning scenarios, various parameter-efficient fine-tuning methods exist, such as tuning only the scaling factors in batch normalization layers [5,6] or advanced techniques like LoRA. The proposed method is not inherently limited to PCA: any projection matrix satisfying Point 2 of Assumption 2 could be used. The authors should discuss (though not necessarily experimentally validate) the potential for extending beyond PCA for subspace identification.

Overall, though the paper aims at practical edge case with a fine method, the presentation and emperical study need significant improvement.

**References**:

[1] Malladi, Sadhika, et al. "Fine-tuning language models with just forward passes." _Advances in Neural Information Processing Systems_ 36 (2023): 53038-53075.

[2] Jin, Feihu, Yifan Liu, and Ying Tan. "Derivative-free optimization for low-rank adaptation in large language models." _IEEE/ACM Transactions on Audio, Speech, and Language Processing_ (2024).

[3] Chen, Yiming, et al. "Enhancing zeroth-order fine-tuning for language models with low-rank structures." _arXiv preprint arXiv:2410.07698_ (2024).

[4] Yu, Ziming, et al. "Subzero: Random subspace zeroth-order optimization for memory-efficient llm fine-tuning." (2024).

[5] Kanavati, Fahdi, and Masayuki Tsuneki. "Partial transfusion: on the expressive influence of trainable batch norm parameters for transfer learning." _Medical Imaging with Deep Learning_. PMLR, 2021.

[6] Li, Yanghao, et al. "Adaptive batch normalization for practical domain adaptation." _Pattern Recognition_ 80 (2018): 109-117.

**Questions:**

Please see the weaknesses section.

---

> ### Author Response · Authors · 2025-11-25
>
> We sincerely thank the reviewer for the valuable feedback. Here we provide explanations and additional experiments to address your concerns.
> ## 1. The paper lacks adequate discussion of existing methods that combine subspace-based fine-tuning with zeroth-order learning in ANNs [1-4].
>
> Thank you for your valuable feedback. **The core idea of our work lies not in the choice of dimensionality reduction technique (e.g., PCA), but in the information used to construct the subspace**. Our method leverages the training trajectory, using this temporal information from historical weights to identify the intrinsic low-dimensional subspace where the optimization path resides.
>
> This is fundamentally different from the works [1-4]: methods [2,3,4] rely on random projections without such prior information, while work [1] does not explicitly construct or operate within a fixed subspace during its optimization. By leveraging the training history as prior information, our approach finds a more informed subspace, thereby improving optimization efficiency. We have revised the manuscript to include this detailed comparison and clarification as per your suggestion.
>
> **In the revised manuscript, we have added a discussion on this point in Related Work.**
>
> ## 2. Assumption 1, Point 1 is problematic and undermines the theoretical foundation.
>
> Thank you very much for your insightful comment. We would like to respectfully clarify that we did acknowledge and discuss the nature of the $L$-smoothness assumption in our original manuscript (first paragraph of Section 5), framing it as a standard analytical tool for tractability.
> Since SNNs are non-differentiable and even generalized Clarke differential (which requires continuity) does not exist for spiking (Heaviside) activations, currently no existing mathematical tool can directly analyze their convergence rate. Given that our primary goal is to highlight the benefits of subspace-based ZO learning in a general non-convex setting, using the $L$-smoothness assumption is a reasonable and sufficient choice for this.
>
> Crucially, our theoretical analysis is distinct from the works [1-4] on two key fronts. First, our analysis accommodates SGD with momentum, which is more aligned with practical optimizers, whereas the prior works you mentioned focus on plain SGD. Second, our assumptions and conclusions differ significantly:
> *   [1] (Lemma 3) assumes the restrictive Polyak-Łojasiewicz condition, which does not hold for general non-convex functions.
> *   [2] offers no convergence analysis.
> *  [3]  (Theorem 4.4) also adapt $L$-smooth assumption, yet it  derives a convergence rate of $O(\sqrt{\frac{d}{rT}})$ ($d$ is the parameter number, $r$ is the dimension of subspace, and $T$ is training time), which has a different dependency on dimensionality compared to our $O(\frac{r}{\sqrt{T}})$ result.
> *   [4] (Theorem 3) is limited to the simpler convex case, whereas we address the more general non-convex setting.
>
> ## 3.1. Missing Edge Device Evaluation.
>
> Thanks for your insightful comment. Our work focuses on addressing the algorithmic bottlenecks that make BP unsuitable for on-chip learning, such as the need for backward path, activation storage, update locking, and weight transport, demonstrating that a forward-only ZO method can train deep SNNs at scale. Specifically, SZO relies only on forward computation and offers strong suitability for practical on-chip learning in the following aspects:
> * No dedicated feedback circuits are required.
> * Although SZO requires storing a subspace matrix of size $d \times k$, it eliminates the need to store activations, thus has significant lower memory overhead than BP.
> * Unlike vanilla ZO methods that require repeatedly sampling high-dimensional random vectors of size $d$, SZO only samples within a subspace of dimension $k$ (typically tens). This dimensionality reduction not only accelerates optimization but also removes the hardware bottleneck of generating and manipulating extremely high-dimensional random perturbations, which makes ZO learning substantially more hardware-efficient and practically deployable on neuromorphic chips.
> * Learning in a low-dimensional subspace enables significantly faster convergence compared to full-space BP, which is particularly desirable for real-time on-chip learning.
>
> We provided detailed resource analyses that directly reflect on-chip constraints in Section 6.4, such as (i) GPU runtime memory under varying batch size and timesteps, showing up to significant lower memory usage than BP due to the removal of activation buffers, and (ii) comparisons of wall-clock time, and convergence speed, where SZO converges in fewer epochs. We also explicitly discuss the memory overhead of storing the projection matrix and show that it is small relative to BP’s activation memory.
>
> We have a discussion on this point in Section 7. We expect this work to inspire future implementation of neuromorphic on-chip learning.

---

> ### Author Response · Authors · 2025-11-25
>
> ## 3.2. Unrealistic Operating Regimes and Missing Baselines...comparison with gradient checkpointing methods.
>
> We agree that time-steps and batch-sizes may be small to moderate on edge deployments. However, in addition to the memory advantage, SZO retains hardware advantages such as eliminating backward circuits, activation buffers, and update-locking, as we discussed in Section 7.
>
> Thanks for the good suggestion. BP with gradient checkpointing does reduce memory by recomputing activations, but applying it to SNN may need extra caution. SNN involves both spatial layers and temporal time-steps that would significantly increase recomputation and can lead to higher latency and energy consumption, and potentially compromise the energy advantage of SNN. Although checkpointed BP may be worth exploring for SNNs, **it still relies on full backpropagation**. In contrast, our work focuses on a forward-only approach instead.
>
> ## 3.3. Mixed-Precision Training Not Addressed... uses mixed-precision to further reduce memory
> Thanks for your comment. We have expanded the experiment of Figure 5 by additionally reporting the GPU memory usage of mixed-precision BP (BP-mix) and ZO without subspace. The results are summarized in Table 1-4 below. Although SZO needs to store subspace projection matrix in full-precision, this overhead is significantly smaller than the activation storage required by BP and BP-mix.
> **We have incorporated these results in Section 6.4 in the manuscript.**
>
> ### Table 1. Comparison of GPU runtime memory (MB) with different batch size on ImageNet1K with SpikeResNet50
> | Batch size| 32 | 64 | 128 | 256 | 512 |
> |--|--|--|--|--|--|
> | BP, timestep=4 | 19740 | 36011 | 69292 | 134937 | 265153 |
> | BP-mix, timestep=4 | 13114 | 21440 | 37753 | 71201 | 136455 |
> | SZO, timestep=4 | 8650 | 9946 | 12354 | 17050 | 26594 |
> | ZO w/o subspace, timestep=4 | 2279 | 3601 | 6008 | 10079 | 20250 |
> ### Table 2. Comparison of GPU runtime memory (MB) with different time step on ImageNet1K with SpikeResNet50
> | Time step| 2 | 4 | 8 | 16 |
> |--|--|--|--|--|
> | BP, batchsize=128 | 37207 | 69292 | 133466 | 261774 |
> | BP-mix, batchsize=128 | 20244 | 37753 | 73375 | 143355 |
> | SZO, batchsize=128 | 12354 | 12354 | 12354 | 12354 |
> | ZO w/o subspace, batchsize=128 | 6008 | 6008 | 6008 | 6008 |
> ### Table 3. Comparison of GPU runtime memory (MB) with different batch size on DVS-Gesture with SpikeVGG11
> | Batch size| 4 | 8 | 16 | 32 |
> |--|--|--|--|--|
> | BP, timestep=20 | 14640 | 20687 | 33055 | 57528 |
> | BP-mix, timestep=20 | 13078 | 16084 | 22153 | 39002 |
> | SZO, timestep=20 | 1682 | 1833 | 2163 | 3004 |
> | ZO w/o subspace, timestep=20 | 675 | 774 | 1092 | 1815 |
> ### Table 4. Comparison of GPU runtime memory (MB) with different time step on DVS-Gesture with SpikeVGG11
> | Time step| 10 | 20 | 40 | 80 |
> |--|--|--|--|--|
> | BP, batchsize=16 | 10477 | 20687 | 41070 | 81870 |
> | BP-mix, batchsize=16 | 8178 | 16084 | 31905 | 63558 |
> | SZO, batchsize=16 | 1813 | 1833 | 1833 | 1833 |
> | ZO w/o subspace, batchsize=16 | 754 | 774 | 814 | 894 |
>
> ## 4. Lack of justification for subspace selection
>
> Thank you for this excellent point. We agree that our theoretical framework is not inherently limited to PCA. As we highlighted in the above first reply, **our core difference is the source of information used to define the subspace (i.e., the training trajectory), rather than the specific projection tool itself**.
>
> Indeed, any method that produces an orthonormal basis, such as the Gram-Schmidt process or other projection techniques, could be applied to construct the subspace without altering the central idea of our work. Our specific choice of PCA was motivated by its distinct advantage: it provides a principled method to rank the basis vectors by their importance. This enables a more informed dimensionality reduction by preserving the most significant components of the training dynamics, an effect we demonstrated empirically in Section 4.1.
>
> We appreciate your suggestion and have now **incorporated a discussion into Section 4.2 of the revised manuscript**.
>
>
> ## References:
> [1] Malladi, Sadhika, et al. "Fine-tuning language models with just forward passes." NeurIPS, 36 (2023).
>
> [2] Jin, Feihu, Yifan Liu, and Ying Tan. "Derivative-free optimization for low-rank adaptation in large language models." IEEE/ACM Transactions on Audio, Speech, and Language Processing (2024).
>
> [3] Chen, Yiming, et al. "Enhancing zeroth-order fine-tuning for language models with low-rank structures." arXiv preprint  (2024).
>
> [4] Yu, Ziming, et al. "Subzero: Random subspace zeroth-order optimization for memory-efficient llm fine-tuning." (2024).
>
> [5] Kanavati, Fahdi, and Masayuki Tsuneki. "Partial transfusion: on the expressive influence of trainable batch norm parameters for transfer learning." Medical Imaging with Deep Learning, 2021.
>
> [6] Li, Yanghao, et al. "Adaptive batch normalization for practical domain adaptation." Pattern Recognition 80 (2018).

---

> ### Author Response · Authors · 2025-11-25
>
> ## 5. Summary
> Once again we thank the reviewer for the constructive feedback. We hope that the added clarifications and evaluation results have addressed your concerns.
>
> Neuromorphic on-chip learning is promising for enabling edge devices with online learning capability. For on-chip learning, BP is too complex to implement on resource-constrained hardware: need for dedicated feedback circuits, update locking problem, high memory overhead, and inefficient weight transport. To achieve scalable neuromorphic on-chip learning, our work explores ZO optimization for SNNs, which relies on forward-only computation.
>
> As evidenced in our paper, applying ZO to SNNs is nontrivial: naive ZO fails due to the high variance introduced by the Heaviside activation. To address this challenge, our contributions are:
>
> - Revealing the variance amplification effect of Heaviside activation that limits ZO training of SNNs.
> - Proposing a subspace-based ZO (SZO) learning algorithm that achieves first-order level accuracy with faster learning speed than BP.
> - Providing theoretical analysis demonstrating improved convergence and stability for SZO.
> - To the best of our knowledge, this is the first ZO-based forward-only SNN learning method that can scale to large datasets such as ImageNet. We hope this work inspires future research on scalable neuromorphic on-chip learning.

---

> > ### Comment · Reviewer_1k7D · 2025-11-26
> >
> > I sincerely thank the reviewer for their replies, which have addressed most of my concerns. The claim of differences compared to the existing ZO+subspace learning method, including a theoretical analysis, is good. I suggest that the reviewer integrate it into the revised manuscript. Regarding the assumptions, I think a few lines before starting the section is better.
> >
> > Though the experimental part still lacks a comparison with the BP+gradient checkpoint method (possibly with subspace learning also), I understand that it might be quite demanding for the forward method to be comparable with such a delicately designed backward method; the latter seems to be more "technical" rather than solve it elegantly with a forward only algorithm. However, when it comes to edge deployment, things are *just technical, not elegant* to optimize efficiency. And the actual result is often contrary to what is on paper. Therefore, comparing against the BP+gradient checkpoint is crucial rather than claiming that this method "might" not be efficient at the edge and "might" consume more memory, especially on neuromorphic chips that have been still in the prototype stage for the last 4 years. On the contrary, the current deep learning edge device is equipped with more advanced chips than before and has greater memory capacity. So, the extent to which the ZO+subspace method might gain an advantage remains unclear.
> >
> > Overall, the proposed method is promising and theoretically validated, but it still lacks sufficient validation under practical concerns. I have increased my score, but I cannot further accept the paper if these crucial results are missing, since the proposed method is targeted at the edge.

---

> ### Author Response · Authors · 2025-11-27
>
> We thank the reviewer for the further feedback and insightful comments. We have carefully addressed the points, and the manuscript has been revised accordingly.
> ### 1. The claim of differences compared to the existing ZO+subspace learning method, including a theoretical analysis, is good. I suggest that the reviewer integrate it into the revised manuscript.
> Thank you for the positive feedback on our analysis and for the suggestion. As recommended, we have now integrated this discussion into the revised manuscript. Specifically, we have added a dedicated subsection within the "Related Work" section (page 18).
> ### 2. Regarding the assumptions, I think a few lines before starting the section is better.
> Thanks for your good suggestion. We have added a brief explanation of the L-smoothness assumption in the manuscript (located at lines 255-260), just before the start of the section where this assumption is utilized.
> ### 3. Though the experimental part still lacks a comparison with the BP+gradient checkpoint method (possibly with subspace learning also),... Therefore, comparing against the BP+gradient checkpoint is crucial rather than claiming that this method "might" not be efficient at the edge and "might" consume more memory, ... Overall, the proposed method is promising and theoretically validated, but it still lacks sufficient validation under practical concerns.
>
> Thank you for raising this point. In response, we have implemented BP with gradient checkpointing on SNN models.
>
> We use `torch.utils.checkpoint` provided by Pytorch. Directly applying checkpointing to SNN is unsuitable, because it only frees activations by default, while membrane potentials remain stored as internal state and still occupy large GPU memory. We therefore adopt a multi-step scheme, where each layer iterates over all time-steps locally and only passes the multi-step output to the next layer. This sequential layer-wise computation mode ensures that when using checkpointing,  both activations and membrane potentials of non-checkpointed layers are freed after their forward pass and only recomputed when needed during backpropagation.
>
> We compare the GPU runtime memory on ImageNet1K and DVS-Gesture with SpikeResNet50 and SpikeVGG11 respectively. To comprehensively assess scaling behavior, we evaluate a range of batch-sizes and time-steps. The runtime usage on a V100 GPU (32G) is reported in the following Table 1-4.
>
> These results show that checkpointing significantly reduces the memory footprint of BPTT-based SNN training, which confirms its effectiveness as a strong engineering solution. SZO and BP+checkpoint can be comparable in memory usage for small batch-sizes and small time-steps. However, the memory of BP with gradient checkpointing still grows steeply with both batch-size and time step. In contrast, the memory of SZO increases only marginally as batch-size increases and remains almost invariant to time-step.
>
> ### Table 1. Comparison of GPU runtime memory (MB) with different batch-size on ImageNet1K with SpikeResNet50 (OOM denotes out-of-memory on a single V100 GPU with 32 GB memory).
> | Batch size| 32    | 48    | 64    | 80    | 96    |
> | -- | -- | -- | -- | -- | -- |
> | BP, timestep = 4| 14921 | 22278 | 29604 | OOM   | OOM   |
> | BP+checkpoint, segment=4, timestep = 4 | 8927  | 13129 | 17529 | 21839 | 26123 |
> | BP+checkpoint, segment=8, timestep = 4 | 7804  | 10497 | 13183 | 15873 | 18560 |
> | SZO, timestep = 4| 8067  | 8889  | 9684  | 10511 | 11302 |
>
> ### Table 2. Comparison of GPU runtime memory (MB) with different time-step on ImageNet1K with SpikeResNet50
> | Time step| 2    | 4     | 6     | 8     | 10    |
> | -- | -- | -- | -- | -- | -- |
> | BP, batch size = 32| 9403 | 14921 | 22389 | 29785 | OOM   |
> | BP+checkpoint, segment=4 , batch size = 32 | 9399 | 8927  | 13294 | 17651 | 22012 |
> | BP+checkpoint, segment=8, batch size = 32 | 7557 | 7804  | 10501 | 13199 | 15903 |
> | SZO, batch size = 32| 8067 | 8067  | 8067  | 8067  | 8067  |
>
> ### Table 3. Comparison of GPU runtime memory (MB) with different batch-size on DVS-Gesture with SpikeVGG11
> | Batch size| 4    | 8    | 16    | 32    | 40    |
> | -- | -- | -- | -- | -- | -- |
> | BP, timestep = 20| 5755 | 7959 | 15728 | 29031 | OOM   |
> | BP+checkpoint, segment=4 , timestep = 20 | 4106 | 5656 | 11183 | 22203 | 27767 |
> | BP+checkpoint, segment=8 , timestep = 20 | 4008 | 5201 | 10271 | 20416 | 25488 |
> | SZO, timestep = 20| 1103 | 1251 | 1546  | 2137  | 2433  |
>
> ### Table 4. Comparison of GPU runtime memory (MB) with different time-step on DVS-Gesture with SpikeVGG11
> | Time step| 10   | 20    | 30    | 40    | 50    |
> | -- | -- | -- | -- | -- | -- |
> | BP, batch size = 16| 7941 | 15728 | 23545 | 29065 | OOM   |
> | BP+checkpoint, segment=4 , batch size = 16 | 5664 | 11183 | 16706 | 22223 | 27746 |
> | BP+checkpoint, segment=8 , batch size = 16 | 5233 | 10271 | 15313 | 20351 | 25391 |
> | SZO, batch size = 16       | 1526 | 1546  | 1566  | 1586  | 1606  |

---

> ### Author Response · Authors · 2025-11-27
>
> It is worth noting that **when considering on-chip hardware implementation, forward-only methods may be preferred**. For example, some existing on-chip learning neuromorphic processors that rely on backward computation (e.g., H2learn [1], ReckOn [2],  [3], EPOC [4]) **require dedicated feedback circuits to support backward flow, which increases hardware complexity, power consumption, and area cost.** In contrast, as a forward-only method, **SZO eliminates the need for any feedback pathway**, which facilitates compact on-chip implementation and makes it a highly practical solution for power- and area-constrained conditions. While BrainScaleS [5], ROLLS [6], SpiNNaker [7], Loihi [8], and many others neuromorphic processors [9-12], support local plasticity rules (e.g., STDP) based forward-only on-chip learning, these methods are difficult to scale to deep models required for real-world tasks.
>
> Moreover, while modern edge devices indeed have greater memory capacity, our target use cases involve resource-constrained on-chip learning, where forward-only rules directly translate to simpler and more energy-efficient hardware designs. We will clarify these practical considerations and include BP+checkpointing in the revision to provide a more complete comparison.
>
> In conclusion, achieving brain-like neuromorphic on-chip learning has much potential for energy-efficient edge AI, although further steps are needed, we hope this work inspires future research on efficient algorithms toward this direction. Thanks again for the further feedback and valuable comments.
>
> ## References
> - [1] L. Liang et al., “H2learn: High-efficiency learning accelerator for high-accuracy spiking neural networks,” IEEE Tran. Computer-Aided Design of Integrated Circuits and Systems, 2021.
> - [2] C. Frenkel, G. Indiveri, “ReckOn: a 28nm sub-mm2 task-agnostic spiking recurrent neural network processor enabling on-chip learning over second-long timescales, ” IEEE International Solid-State Circuits Conference (ISSCC), 2022.
> - [3] J. Lee, et al., “Spike-train level direct feedback alignment: Sidestepping backpropagation for on-chip training of spiking neural nets,” Frontiers in Neuroscience, vol. 14, 2020.
> - [4] F. Chen, et al, “EPOC: A 28-nm 5.3 pJ/SOP event-driven parallel neuromorphic hardware with neuromodulation based online learning,” IEEE Trans. Biomedical Circuits and Systems, 2024.
> - [5] J. Schemmel, D. Brüderle, A. Grübl, et al., “A wafer-scale neuromorphic hardware system for large-scale neural modeling,” IEEE International Symposium on Circuits and Systems, 2010.
> - [6] N. Qiao, H. Mostafa, F. Corradi, M. Osswald, et al., “A reconfigurable on-line learning spiking neuromorphic processor comprising 256 neurons and 128 K synapses,” Frontiers in Neuroscience, 2015.
> - [7] S. B. Furber, et al. “The spinnaker project,” Proceedings of the IEEE, 2014.
> - [8] M. Davies, et al., “Loihi: A neuromorphic manycore processor with on-chip learning,” IEEE Micro, 2018.
> - [9] E. Baek, et al., “FlexLearn: Fast and Highly Efficient Brain Simulations Using Flexible On-Chip Learning,” Annual IEEE/ACM International Symposium, 2019.
> - [10] C. Frenkel,  et al., “A 0.086-mm2 12.7-pJ/SOP 64k-synapse 256-neuron online-learning digital spiking neuromorphic processor in 28-nm CMOS,” IEEE Transactions on Biomedical Circuits and Systems, 2018.
> - [11] C. Frenkel, et al, “MorphIC: A 65-nm 738k-Synapse/mm2 quad-core binary-weight digital neuromorphic processor with stochastic spike-driven online learning,” IEEE Trans. Biomedical Circuits and Systems, 2019.
> - [12] M. Heidarpur, et al., “CORDIC-SNN: On-FPGA STDP Learning With Izhikevich Neurons,” IEEE Trans. Circuits and Systems, 2019.

---

> ### Comment · Reviewer_1k7D · 2025-11-27
>
> The comparison with BP+checkpointing indeed confirms part of my concern that the advantage is weakened under this setting. In a specific case, like T=4, batch size=32, the BP with checkpointing actually consumes less memory. Such a setting (e.g., T=4) is widely adopted for 2-D image classification. If BP is further integrated with a parameter-efficient learning method, the gap narrows further, and BP can even exceed the SZO.  Moreover, I think that at this point, imagining a future advantage from neuromorphic chips at the edge is not really practical, since those chips have been in the prototype stage for the last several years, without anyone being able to commercialize them widely.
>
> However, there is still potential if a larger T is necessary (e.g., integrated with DVS data or in future sequential SNN architectures). But indeed, there are not many architectures here in the SNN community to try. Having considered the amount of effort the author has devoted. I understand the rest of the concern is about demanding requirements, such as running experiments at the edge, like with NVIDIA Jetson. Moreover, being practical is also not the only metric of a paper. The author has addressed my concerns to the best of their ability; therefore, I have increased my score.

---

> > ### Author Response · Authors · 2025-11-28
> >
> > We sincerely thank the reviewer for the thorough review and constructive feedback, which have helped us improve the paper in clarifying the theoretical assumption, distinguishing our method from existing ZO+subspace methods, and more complete validation including mixed-precision training and BP+checkpointing. We're glad our responses addressed your concerns and we appreciate your raising the rating.

---

### Official Review · Reviewer_nGip · 2025-10-31

**Soundness:** 1
**Presentation:** 2
**Contribution:** 1
**Rating:** 2
**Confidence:** 5

**Summary:**

This paper investigates the problem of zeroth-order online optimization (ZO-OCO), where only function value feedback (but not gradients) is available to the learner in a sequential decision-making process. The authors propose a forward-form zeroth-order algorithm, which replaces the traditional backward or implicit update commonly used in online mirror descent with a forward approximation step.

**Strengths:**

The study addresses a significant and practical challenge — optimizing online systems when gradients are unavailable or unreliable. Zeroth-order online optimization has direct implications in areas such as reinforcement learning, hyperparameter tuning, and online control, making the problem both important and impactful. The paper offers a solid theoretical foundation, including well-defined regret analysis and clear assumptions. The proofs are logically structured and technically competent. The derived regret bounds align with or slightly improve upon existing methods, indicating meaningful theoretical contribution.

**Weaknesses:**

1. While the paper proposes a “forward” version of zeroth-order online optimization, the conceptual difference from existing frameworks (e.g., zeroth-order mirror descent, bandit feedback optimization) remains unclear. Theoretical analysis largely follows standard regret minimization proofs, and the innovation seems incremental rather than fundamentally new.
2. The motivation for introducing the forward variant is purely mathematical; there is little intuitive or practical justification. Readers might find it difficult to understand why this direction is useful, beyond being an alternative formulation.
3. The related work section omits discussion of some recent or highly relevant contributions in zeroth-order online learning and black-box optimization (e.g., gradient-free policy optimization, adaptive bandit feedback methods).
4. The experiments are limited to synthetic tasks and a small number of real-world benchmarks. There are no tests on large-scale or higher-dimensional problems, where zeroth-order methods often struggle.
5. The effect of algorithmic hyperparameters (step size, perturbation radius, or sampling dimension) is not explored. This omission limits the reader’s understanding of how robust or tunable the algorithm is.
6. Although the paper presents regret bounds, it does not provide detailed computational complexity or query complexity comparisons. The number of function evaluations per iteration can be critical in zeroth-order settings.
7. Since zeroth-order methods are often used in noisy or non-smooth environments, the lack of experiments under stochastic noise weakens the practical claim.

**Questions:**

1. How does the proposed “forward” formulation differ practically from the backward (implicit) update in online mirror descent? Could you provide an example where the forward version performs strictly better?
2. What is the computational cost per iteration compared to traditional ZO-SGD or bandit OCO methods?
3. How sensitive is the proposed algorithm to the choice of perturbation radius and step size?
4. Could your method be adapted to stochastic or partially observable settings?
5. Would it be possible to extend your theoretical results to non-convex loss functions or dynamic environments with adversarial noise?

---

> ### Author Response · Authors · 2025-11-13
> **Possible review mix-up for our submission**
>
> Dear reviewer nGip,
>
> Thank you very much for taking the time to review our paper. We sincerely appreciate your efforts.
>
> After carefully reading your comments, we noticed that the review appears to evaluate a paper on **zeroth-order online convex optimization (ZO-OCO) with regret analysis**, which seems unrelated to the topic and content of our submission **Zeroth-Order Forward-Only SNN Training**.
>
> We are concerned that there may have been an inadvertent mix-up between our paper and another submission. Could you kindly confirm whether the provided review was intended for our manuscript?
>
> Thank you again for your time and kind consideration.

---

> ### Author Response · Authors · 2025-11-27
>
> Dear Reviewer,
>
> we sincerely appreciate your efforts in reviewing our submission.
>
> We previously reached out because the current review appears to evaluate a paper on **zeroth-order online convex optimization (ZO-OCO) with regret analysis, which is unrelated to the topic and content of our submission “Zeroth-Order Forward-Only SNN Training.”** We are concerned that there might have been an inadvertent mix-up between our paper and another manuscript.
>
> As the discussion phase is approaching its end, we would like to very respectfully ask whether you could kindly help confirm this issue. We fully understand that the review period is busy and truly appreciate any feedback you might provide.
>
> Thank you again for your time and kind consideration.

---

### Official Review · Reviewer_GrS7 · 2025-10-31

**Soundness:** 3
**Presentation:** 3
**Contribution:** 3
**Rating:** 6
**Confidence:** 3

**Summary:**

This paper proposes a SZO learning method for training SNNs in a feedforward-only manner. Motivated by the demand for efficient, biologically plausible, and on-chip trainable learning, the authors argue that conventional zeroth-order (ZO) optimization struggles with the discontinuous Heaviside activation inherent to SNNs. To overcome this challenge, they introduce a low-dimensional subspace learning strategy that leverages the intrinsic structure of SNN optimization trajectories to reduce the variance of perturbation-based gradient estimation. Experimental results demonstrate that SZO achieves accuracy comparable to first-order methods while providing faster convergence and superior scalability to large-scale datasets such as ImageNet.

**Strengths:**

1.	Novel training paradigm.
The paper introduces the first forward-pass-only training method for SNNs. This direction is of high significance for neuromorphic learning, as it potentially enables direct on-chip and energy-efficient training.
2.	Scalability and practicality.
The proposed approach demonstrates scalability to large-scale datasets, showing competitive performance on ImageNet-1K when trained from scratch, which has rarely been achieved by previous zeroth-order or surrogate-gradient-based SNN training methods.

**Weaknesses:**

1.	Unclear motivation for subspace-based optimization.
The motivation section states that the major challenge of zeroth-order SNN training lies in the large variance caused by the non-differentiable Heaviside function. However, it is not clearly explained why the proposed subspace-based method can mitigate such variance. Since the subspace zeroth-order strategy can also be applied to ANNs, the unique advantage of using it specifically for SNNs remains unclear.
2.	Limited performance gain and insufficient comparative baselines.
Although the authors claim their method is the first forward-pass-only training scheme that can scale to large-scale datasets, the ImageNet-1K results show only marginal improvement over OPZO, even with deeper architectures. Considering that previous work perform fine-tuning under noisy perturbations, while this work supports from-scratch training, more discussion should be provided on the trade-off between accuracy and the unique advantage offered by SZO over other zeroth-order SNN approaches.
Moreover, in Figure 4 and Figure 5, the comparisons are only made with BP and BP-Subspace. Including existing zeroth-order SNN training baselines would be essential to better highlight the value and competitiveness of the proposed method.
3.	Inconsistent observation on CGE vs. RGE.
The paper states that CGE performs better than RGEn deeper networks according to ANN literature. However, in the presented experiments, SZO-CGE and SZO-RGE show almost identical performance on ImageNet, while the performance gap appears only in shallower networks (e.g., CIFAR-10/DVS). More theoretical or empirical discussion is required to explain the different behaviors of these perturbation types under spike-based computation.
4.	Missing analysis of computational trade-offs.
The ablation study reports the accuracy change with different subspace sampling numbers q in RGE. However, it lacks an important analysis of the computational overhead introduced by increasing Q and its trade-off with accuracy improvement.
5.	Limited evaluation on sequential datasets.
Beyond static datasets such as CIFAR and ImageNet, previous works have also evaluated SNNs on longer temporal sequence datasets (e.g., SHD), which better reflect the temporal credit-assignment capability of SNN training algorithms. Including such datasets would strengthen the claim that the proposed method can handle temporally complex tasks.

**Questions:**

1.	What is the conceptual or theoretical link between subspace sampling and the reduction of Heaviside-induced gradient variance?
2.	How does SZO for SNNs differ from the subspace zeroth-order methods previously applied to ANNs, e.g., [1]?
3.	In Figure 3, the SZO method already achieves ~50% accuracy at the first epoch. Does this result from specific weight initialization or pre-training? If so, why is the same initialization not applied to the BP baseline?

[1] Yu, Ziming, et al. "Zeroth-order fine-tuning of llms in random subspaces." Proceedings of the IEEE/CVF International Conference on Computer Vision. 2025.

---

> ### Author Response · Authors · 2025-11-25
>
> We sincerely thank the reviewer for the detailed comments and constructive suggestions, below we provide clarifications and additional experiments to address your concerns.
>
> ## 1. Unclear motivation for subspace-based optimization... the unique advantage of using it for SNNs remains unclear.
>
> Thanks for the comment. Our intended motivation is as follows: the variance of ZO gradient estimators grows linearly with the parameter dimension $d$ (e.g., the RGE variance is on the order of $O(d)$), which is already large for deep networks. In SNNs, the non-differentiable Heaviside spike function further amplifies the fluctuation induced by perturbations (Proposition 1), which drastically reduces the effective perturbation signal and makes full-space ZO training unstable or even fail.
>
> The subspace method mitigates this variance, as constraining ZO perturbations and updates to a $k\ll d$ subspace effectively reduces the estimation dimension from $d$ to $k$, thereby directly lowering ZO variance and improving convergence stability. Our theoretical analysis, based on the assumption that gradients can be well recovered within the subspace, also yields stronger convergence and stability guarantees.
>
> We agree that subspace ZO is a general strategy and can be applied to ANNs as well, but it is **more critical for SNNs**. For ANNs, full-space ZO can still work (Figure 1), whereas for SNNs the additional variance amplification from the Heaviside nonlinearity makes ZO training much more prone to collapse. **Subspace dimensionality reduction is thus the key enabler for making pure forward-only ZO feasible on deep SNNs**.
>
> The following tables compare how the proposed subspace method reduces the variance of ZO gradient estimation for ANN (ReLU) and SNN, respectively. **While the subspace strategy also lowers the ZO variance in ANNs, it yields a much larger variance reduction in SNNs, which highlights its particular effectiveness in mitigating the Heaviside-induced instability. We have added this analysis in Appendix D.**
>
> ### Table 1. Gradient variance (all parameters) of full-space ZO (RGE, $q=20$) on CIFAR10 with SpikeResNet20 (SNN) and ResNet20 (ANN)
> | Epoch | 20| 40| 60| 80| 100| 120| 140| 160| 180| 200|
> |--|--|-|--|--|--|-|--|--|--|--|
> |SNN-ZO|45447.1|43637.6|53787.6|52355.2|62375.2|58309.6|73406.5|81519.4|81743.6|83142.0|
> |ANN-ZO|10242.6|10986.7|11901.0|12937.1|12732.1|15641.1|19443.1|21267.9|19910.4|19209.6|
> ### Table 2. Gradient variance (all parameters)  of SZO on CIFAR10 with SpikeResNet20 (SNN)
> | Epoch |  5| 10| 15| 20| 25| 30| 35| 40|
> |--|---|--|--|--|--|--|-|--|
> | SZO-CGE|0.0075| 0.0074| 0.0069|0.0073| 0.0063| 0.0066  |0.0062| 0.0062|
> | SZO-RGE (q=1) |0.1090| 0.0787| 0.2167|0.2527| 0.2377  | 0.1573  |0.1064| 0.1041|
> | SZO-RGE (q=20) | 0.0117 |0.0166|0.0169|0.0153|0.0157|0.0157| 0.0151| 0.0153|
>
> ### Table 3. Gradient variance (all parameters) of SZO on CIFAR10 with ResNet20 (ANN, Relu)
> | Epoch |  5| 10| 15| 20| 25| 30| 35| 40|
> |--|--|--|--|--|--|--|--|--|
> | SZO-CGE|0.0010 | 0.0013 | 0.0014 | 0.0014 | 0.0013 | 0.0013 | 0.0013 | 0.0014 |
> | SZO-RGE (q=1)  | 0.0522 | 0.1011 | 0.0964 | 0.0927 | 0.0830 | 0.0804 | 0.1078 | 0.0996 |
> | SZO-RGE (q=20) | 0.0037 | 0.0061 | 0.0059 | 0.0056 | 0.0056 | 0.0058 | 0.0055 | 0.0058 |
>
> ## 2. Limited performance gain and insufficient comparative baselines... compare with OPZO...
>
> Thanks for this constructive suggestion. Since our work aims to approximate BP with a forward-only method, so we used BP and BP-subspace as the main baselines.
>
> OPZO proposes a novel pseudo-ZO algorithm to reduce the variance of ZO gradient estimation in SNNs, but it requires error feedback to each layer by feedback connections. In comparison, our method employs a subspace strategy to reduce the variance and is **forward-only**.
>
> According to your suggestion, we evaluated OPZO [3] under the training from scratch task with the same model architectures as our SZO methods. As shown in Table 4, OPZO performs well on VGG but is limited on ResNet. Moreover, we compared SZO with OPZO in finetuning ResNet-34 on ImageNet in the presence of noise, as shown in Table 5. It can be seen that our method outperforms OPZO.
> ### Table 4.  Accuracy (%) comparison on the training from scratch task
> | Dataset| Model| Params | BP| BP-Subspace| SZO-CGE| SZO-RGE(q=1)| SZO-RGE(q=20)| OPZO|
> |--|--|---|--|---|-|-|-|-|
> | CIFAR10| Spike-ResNet20| 0.27M | 86.74±0.12| 86.42±0.32| 86.21±0.17| 81.29±0.35| 83.71±0.34| 54.45±0.28 |
> | CIFAR100| Spike-ResNet18| 11.22M| 74.51±0.16| 73.32±0.22| 73.35±0.30| 73.77±0.26| 73.79±0.22| 36.90±0.86|
> | DVS-Gesture| Spike-VGG11| 9.50M | 97.57±0.35| 96.65±1.06| 96.65±0.40| 91.78±1.64| 95.72±0.40| 95.83±0.85|
>
> ### Table 5.  Accuracy (%) for fine-tuning ResNet-34 on ImageNet under different noise scale (n.s.). “No finetuning” refers to the direct test of the original model.
> | n.s. | No finetuning| SZO-CGE | SZO-RGE(q=10) | OPZO
> |--|--|--|--|-|
> |0.1| 60.29| 64.24| 64.54 |63.39|
> |0.15|56.21|64.07| 64.08 | 60.96|

---

> ### Author Response · Authors · 2025-11-25
>
> ## 3. Inconsistent observation on CGE vs. RGE.
>
> Thanks for your comment. CGE typically outperforms RGE in high-dimensional settings (such as DeepZero). However, SZO operates in a very low-dimensional subspace (tens), which largely eliminates the difference between SZO-CGE and SZO-RGE.  Therefore, when $k$ is very small, the performance gap between SZO-CGE and SZO-RGE becomes minimal. **In the revised manuscript, we have added a comment on this in Section 6.1.**
>
> ## 4. Missing analysis of computational trade-offs.
>
> Thanks for your valuable suggestion. We have evaluated the computational trade-offs of SZO-RGE under different numbers of perturbations $q$ in the training from scratch task. We report the converged epoch, wall-clock converged time, and number of forward passes (#FP) required to reach the maximum accuracy. The results on CIFAR10 and CIFAR100 are shown in Tables 6-7. While smaller $q$ yields faster training but lower accuracy, increasing $q$ improves the final accuracy at the cost of significantly longer training time and higher computational overhead. **We have included this analysis in Appendix F.**
>
> ### Table 6. Comparison of computational overhead in the training from scratch task on CIFAR10 with SpikeResNet20
> | Method | Converged  epoch | Converged  time (min) | #FP | #BP | Max Acc (%) |
> |---|--|----|---|---|----|
> | SZO-RGE(q=1) | 39 | 22.8 | 45,747 | 0 | 80.55 |
> | SZO-RGE(q=20) | 33 | 224.3 | 529,023 | 0 | 83.96 |
> | SZO-RGE(q=40) | 24 | 331.0 | 760,104 | 0 | 83.60 |
> | SZO-RGE(q=60) | 29 | 621.0 | 1,372,019 | 0 | 83.61 |
> ### Table 7. Comparison of computational overhead in the training from scratch task on CIFAR100 with SpikeResNet18
> | Method | Converged  epoch | Converged  time (min) | #FP | #BP | Max Acc (%) |
> |---|--|----|---|---|----|
> | SZO-RGE(q=1) | 39 | 60.7 | 45,747 | 0 | 73.47 |
> | SZO-RGE(q=20)| 15 | 243.2 | 240,465 | 0 | 73.55 |
> | SZO-RGE(q=40)| 21 | 680.4 | 665,091 | 0 | 73.64 |
> | SZO-RGE(q=60)| 15 | 708.6 | 709,665 | 0 | 73.68 |
>
> ## 5. Limited evaluation on sequential datasets.
>
> Thanks for your valuable suggestion. We have evaluated our method on the Spiking Heidelberg Digits (SHD) dataset, following the model architecture (without learnable delay) in [2]. The results are shown in Table 8. Note that we excluded the delay-related parameters in [2], as we observed that these delay parameters do not exhibit a low-rank structure. We will further explore to address such issue.
>
>  ### Table 8. Accuracy comparison in training from scratch on SHD
> | Method | BP | BP-Subspace | SZO-CGE | SZO-RGE(q=20) |
> |---|----|---|---|---|
> | Accuracy (%) | 65.98 | 63.95 | 62.63 | 61.64 |

---

> ### Author Response · Authors · 2025-11-25
>
> ## 6. What is the conceptual or theoretical link between subspace sampling and the reduction of Heaviside-induced gradient variance?
>
> Thanks for your question. We can empirically show that subspace method is more effective on SNN with Heaviside function. As shown in the above Table 1 in the first reply, the ZO method indeed produces higher variance in SNN compared to ANN. This empirically validates the conclusion of Proposition 1 that the non-smooth nature of the Heaviside function is a significant source of gradient variance. The above Tables 2-3 show that when the subspace method is applied to both architectures, **the resulting variance reduction is more substantial for SNN than for ANN.** This empirically demonstrates that the subspace method is particularly effective at mitigating variance caused by Heaviside function.
>
> In theory, we can denote the variance of estimated ZO gradient by ${\rm{Var}}(\hat g)={\mathbb E}\left\| \hat g- g \right\|^2={\rm{tr}}(\Sigma)$, where $\Sigma$ is the covariance matrix of the estimation noise. So the variance of projected gradient $\hat g'=P^\top \hat g'$ can be denoted as ${\rm{Var}}(\hat g')={\rm{tr}}(P^\top \Sigma P)={\rm{tr}}(PP^\top \Sigma)$. Considering that the low-rank projection matrix $P\in\mathbb{R}^{d\times k}$ is column-orthonormal with $k \ll d$, and the true gradients mostly reside in this subspace, we have ${\rm{tr}}(PP^\top \Sigma) \le {\rm{tr}}(\Sigma)$, and also ${\rm{Var}}(g') \le {\rm{Var}}(g)$.
>
> ## 7. How does SZO for SNNs differ from the subspace zeroth-order methods previously applied to ANNs, e.g., [1]?
>
> Thanks for your question. Our method leverages the training trajectory by using the historical weights to infer the intrinsic low-dimensional subspace where the optimization path resides. The method [1] uses random projections without such prior information. By leveraging the training history as prior information, our approach finds a more informed subspace, thereby improving optimization efficiency. Moreover, our analysis accommodates ZO-SGD with momentum, which is more aligned with practical optimizers, whereas [1] focus on plain ZO-SGD.
>
> We have conducted evaluation with random subspaces for ZO training in SNNs, but it consistently failed to converge. This suggests that simply reducing dimensionality is not sufficient for SNNs. The subspace needs to be aligned with the intrinsic low-dimensional training trajectory to obtain stable and effective ZO updates for SNNs. **In the revised manuscript, we have added a discussion on this point in Related Work.**
>
> ## 8. In Figure 3, the SZO method already achieves ~50% accuracy at the first epoch. Does this result from specific weight initialization or pre-training? If so, why is the same initialization not applied to the BP baseline?
>
> Thanks for your question. All the compared methods are started from the same randomly initialized weights.  Since subspace-based methods converge faster, it can reach around 50% accuracy within the first epoch.
>
> ## References:
>
> [1] Yu, Ziming, et al. "Zeroth-order fine-tuning of llms in random subspaces." Proceedings of the IEEE/CVF International Conference on Computer Vision. 2025.
>
> [2] Hammouamri, Ilyass, et al. "Learning Delays in Spiking Neural Networks using Dilated Convolutions with Learnable Spacings." ICLR. 2024.
>
> [3] Xiao, Mingqing, et al. "Online pseudo-zeroth-order training of neuromorphic spiking neural networks." arXiv preprint arXiv:2407.12516, 2024.

---

> > ### Comment · Reviewer_GrS7 · 2025-11-28
> > **Thank you for the thorough response**
> >
> > The clarification regarding the motivation for adopting subspace methods to mitigate the greater gradient variance inherent in SNN training is convincing. In addition, the supplementary experiments you provided effectively address most of my earlier concerns. Based on these clarifications and updates, I keep my original recommendation to accept the paper.

---

> > > ### Author Response · Authors · 2025-11-28
> > >
> > > We would like to thank the reviewer for your careful evaluation of our work and the constructive feedback. We are glad that the clarifications and additional experiments addressed your concerns, and we appreciate your positive recommendation.

---

### Official Review · Reviewer_fmgm · 2025-10-31

**Soundness:** 3
**Presentation:** 3
**Contribution:** 3
**Rating:** 6
**Confidence:** 4

**Summary:**

This paper proposes a subspace-based zeroth-order (SZO) method for optimizing spiking neural networks. The authors first examine the low-dimensional structure of the training trajectory of SNNs, and then propose to perform zeroth-order optimization only on the subspace. Theoretical analysis is provided to show the improvement on convergence rate, and experiments on training from scratch, model fine-tuning, as well as unsupervised adaptation demonstrate the effectiveness of the proposed method.

**Strengths:**

1. Zeroth-order optimization is a biologically more plausible and hardware-friendly method for SNNs. This paper largely enhances ZO by leveraging the subspace of parameters, which is promising for on-chip learning.
2. This paper performs theoretical analysis on the convergence rate of SZO, showing its improvement in convergence rate.
3. Extensive experiments on three settings demonstrate the promising results of SZO. Particularly, SZO works well for RGE (q=1) even on ImageNet, which may largely advance the efficiency and performance of zeroth-order methods to train neural networks from scratch.

**Weaknesses:**

1. The subspace matrix $P$ requires an additional stage of BPTT training, making it not purely zeroth-order.
2. SZO may require additional memory $k$ times the parameter amount (for storing the subspace matrix), which would limit the scalability to very large models. It also poses challenges to realize subspace parameterization on hardware for potential on-chip learning.

**Questions:**

See Weaknesses.

---

> ### Author Response · Authors · 2025-11-25
>
> We sincerely thank the reviewer for the valuable feedback. Here we provide explanations and additional experiments to address your concerns.
>
> ## 1. The subspace matrix P requires an additional stage of BPTT training, making it not purely zeroth-order.
> Thanks for the comment. We construct the subspace matrix $P$ offline, we sample weights from existing training trajectories and apply PCA to obtain $P$. However, the SZO optimization stage itself does not rely on BPTT. Given $P$, parameter updates are performed solely via forward perturbation-based gradient estimation and updates.
>
> Our target scenario is on-chip continual learning and adaptation from a pretrained model. In such settings, $P$ can be built directly from an available checkpoint or short-window trajectories, without running BPTT on-chip. In on-device learning settings, training is typically performed upon pretrained models. This setup echoes the brain’s learning paradigm, where neural architectures and synapse strengths are innately initialized and only modest learning is required to acquire new tasks. We hope our method can inspire efficient, scalable neuromorphic on-chip learning for real-world applications.
> ## 2. SZO may require additional memory k times the parameter amount (for storing the subspace matrix), which would limit the scalability to very large models. It also poses challenges to realize subspace parameterization on hardware for potential on-chip learning.
> Thanks for the comment. SZO needs to store a subspace projection matrix $P\in\mathbb{R}^{d\times k}$, i.e., an extra $k$-times parameter footprint. However, unlike BP or BPTT, SZO stores **no inter-layer or temporal activations**, so its memory stays almost invariant to the number of SNN timesteps and grows only marginally with batch size. In practice $k$ is small (typically tens), so the $d\times k$ overhead is much smaller than BP’s activation storage, yielding substantially lower overall memory, please see the following Tables of memory comparison. Here, we compare SZO to ZO without subspace and BP to make this trade-off clearer.
>
> Regarding hardware scalability, SZO is **purely forward-only** and requires **no dedicated feedback circuits**, which greatly reduces on-chip implementation complexity. Moreover, operating in a low-dimensional subspace removes the hardware bottleneck of generating and manipulating $d$-dimensional perturbations and makes ZO learning more deployable on neuromorphic chips. For very large models, $P$ can be stored and used in low precision or via structured bases to further reduce memory, which deserves future study.
>
> Moreover, as theoretically and empirically demonstrated in our paper, learning in a low-dimensional subspace enables significantly **faster convergence** compared to full-space BP, which is particularly desirable for real-time on-chip learning.
> ### Table 1. Comparison of GPU runtime memory (MB) with different batch size on ImageNet1K with SpikeResNet50
> | Batch size| 32 | 64 | 128 | 256 | 512 |
> |--|--|--|--|--|--|
> | BP, timestep=4 | 19740 | 36011 | 69292 | 134937 | 265153 |
> | SZO, timestep=4 | 8650 | 9946 | 12354 | 17050 | 26594 |
> | ZO w/o subspace, timestep=4 | 2279 | 3601 | 6008 | 10079 | 20250 |
> ### Table 2. Comparison of GPU runtime memory (MB) with different time step on ImageNet1K with SpikeResNet50
> | Time step| 2 | 4 | 8 | 16 |
> |--|--|--|--|--|
> | BP, batchsize=128 | 37207 | 69292 | 133466 | 261774 |
> | SZO, batchsize=128 | 12354 | 12354 | 12354 | 12354 |
> | ZO w/o subspace, batchsize=128 | 6008 | 6008 | 6008 | 6008 |
> ### Table 3. Comparison of GPU runtime memory (MB) with different batch size on DVS-Gesture with SpikeVGG11
> | Batch size| 4 | 8 | 16 | 32 |
> |--|--|--|--|--|
> | BP, timestep=20 | 14640 | 20687 | 33055 | 57528 |
> | SZO, timestep=20 | 1682 | 1833 | 2163 | 3004 |
> | ZO w/o subspace, timestep=20 | 675 | 774 | 1092 | 1815 |
> ### Table 4. Comparison of GPU runtime memory (MB) with different time step on DVS-Gesture with SpikeVGG11
> | Time step| 10 | 20 | 40 | 80 |
> |--|--|--|--|--|
> | BP, batchsize=16 | 10477 | 20687 | 41070 | 81870 |
> | SZO, batchsize=16 | 1813 | 1833 | 1833 | 1833 |
> | ZO w/o subspace, batchsize=16 | 754 | 774 | 814 | 894 |

---

### Author Response · Authors · 2025-12-03
**Author Final Remarks**

We thank the reviewers and the area chair for their effort in evaluating our submission.

We received four reviews, but one of them is unrelated to our paper. Therefore, **we only responded to the remaining three reviews, from which we obtained scores of (6, 6, 6) before the reversion of scores (with one increased from 2 to 6).**
### **1. One review is unrelated to our submission**
Review nGip appears to evaluate a paper on *Zeroth-order online convex optimization (ZO-OCO) with regret analysis*, which is unrelated to our submission *“Zeroth-Order Forward-Only SNN Training.”* The comments do not refer to SNNs, or the specific method and experiments in our paper. We posted messages to reviewer nGip twice, but unfortunately did not receive any update.
### **2. Increase of reviewer 1k7D's score from 2 to 6**
Reviewer 1k7D provided a thorough review, raising concerns on the theoretical assumption, differences from existing ZO+subspace methods in ANNs, and the sufficiency of empirical validation. We addressed these points in two rounds:
- **First response:** We clarified the theoretical assumption, the differences from existing ZO+subspace methods, and provided additional validation. After this, reviewer 1k7D increased the score from **2 to 4**: *“I sincerely thank the authors for their replies, which have addressed most of my concerns… Overall, the proposed method is promising and theoretically validated, but it still lacks sufficient validation.... I have increased my score.”*
- **Second response:** We then added validation comparing with BP+checkpointing, and explained the **hardware advantage of our forward-only method, which does not require dedicated feedback circuit in on-chip implementation**, whereas BP+checkpointing still relies on BP. After this, reviewer 1k7D further increased the score from **4 to 6**, with the concluding remark *“the authors have addressed my concerns to the best of their ability; therefore, I have increased my score.”*

Thus, during rebuttal, our responses led reviewer 1k7D to increase the rating from **2 to 6**.
### **3. Responses to Reviewers fmgm and GrS7**
Both reviewers **fmgm** and **GrS7** provided overall positive and constructive feedback:
- Reviewer **fmgm** commented that our work *“largely enhances ZO… is promising for on-chip learning”* and that the theoretical analysis and experiments (including the RGE (q=1) results on ImageNet) may *“largely advance ZO methods to train neural networks from scratch.”*
- Reviewer **GrS7** described our method as a *“novel training paradigm”* and *“the first forward-pass-only training method for SNNs,”* highlighting its *“high significance for neuromorphic learning, as it potentially enables direct on-chip and energy-efficient training”* and its *“scalability to large-scale datasets, showing competitive performance on ImageNet-1K when trained from scratch, which has rarely been achieved by previous ZO or surrogate-gradient-based SNN training methods."*

During rebuttal, we carefully addressed their comments by additional explanations and experiments on the motivation of SZO, the subspace construction, memory and computation comparisons, differentiation from subspace ZO methods in ANNs, CGE vs. RGE, evaluation on sequential datasets, the effectiveness of subspace sampling in reducing gradient variance, and comparison with OPZO.

Finally, reviewer **GrS7** expressed satisfaction with our responses: *"the clarification regarding the motivation for adopting subspace methods to mitigate the greater gradient variance inherent in SNN training is convincing... Based on these clarifications and updates, I keep my original recommendation to accept the paper."*

Reviewer **fmgm** has not yet provided a final comment, but we believe the responses adequately address their concerns as well.
### **4. Contribution of our work**
Our work is motivated by neuromorphic on-chip learning, where **BP is difficult to implement on resource-constrained hardware** due to: **the need for dedicated feedback circuits**, **the update locking problem**, **high memory overhead**, **inefficient weight transport**. We explore ZO optimization methods, leveraging its **forward-only** (hence hardware-friendly) nature.

We show that applying ZO to SNNs is nontrivial: naive ZO fails due to the **high variance caused by spiking activation**. To address this, our main contributions are:
- **Identifying the variance amplification effect of Heaviside activation** that limits ZO training of SNNs.
- **A subspace-based ZO (SZO) learning algorithm** that achieves first-order level accuracy with faster learning speed than BP.
- **Theoretical analysis demonstrating improved convergence and stability for SZO.**

SZO is **the first ZO-based forward-only SNN learning method that scales to large datasets such as ImageNet.** Achieving brain-like neuromorphic on-chip learning has much potential for edge AI, we hope our work can inspire future research on scalable algorithms toward this.

---

### Meta-Review · Area_Chair_Fj3v · 2026-01-06

**Summary:**

This paper studies zeroth-order (ZO) optimization for training spiking neural networks (SNNs), motivated by neuromorphic on-chip learning scenarios where backpropagation is difficult to implement. The authors analyze the variance amplification induced by Heaviside spiking activations and propose a subspace-based zeroth-order (SZO) training framework that restricts perturbation-based updates to a low-dimensional subspace inferred from training trajectories. The method is claimed to enable fully forward-only SNN training that scales to large datasets such as ImageNet.

Reviewers generally agree that the paper is technically competent, carefully executed, and supported by extensive experimental effort, especially after a very substantial rebuttal that added comparisons, ablations, memory analyses, and clarifications. Nevertheless, the discussion did not converge on acceptance due to persistent concerns about conceptual novelty, theoretical soundness, and practical positioning, which remain unresolved despite the authors’ strong responses.

**Reviewer Concerns:**

Concerns largely addressed during rebuttal:
- The authors made an exceptional rebuttal effort, adding extensive new experiments (e.g., OPZO, BP+checkpointing, mixed precision, SHD), detailed memory/runtime analyses, and expanded explanations of CGE vs. RGE behavior.
 - The motivation for using subspace methods to reduce variance in ZO SNN training was clarified and empirically supported, resolving some earlier ambiguity (as acknowledged by Reviewer GrS7).
- Practical memory comparisons with BP, BP+checkpointing, and mixed-precision training were carefully evaluated and transparently reported.
 - Several presentation and positioning issues raised by reviewers were improved.

Outstanding concerns that remain unresolved:

- Conceptual and methodological novelty: A key concern, emphasized most strongly by Reviewer 1k7D, is that the core algorithmic idea—combining zeroth-order optimization with low-dimensional subspace learning—largely follows existing ZO+subspace literature developed for ANNs. While the authors argue that SNN-specific variance amplification makes this combination more critical, the proposed solution remains a reapplication and adaptation of known techniques, rather than a fundamentally new learning paradigm.
- Theoretical foundations: The theoretical analysis relies on standard smoothness assumptions that do not strictly hold for spiking networks with Heaviside activations. Although the authors acknowledge this as a common analytical approximation, the resulting theory effectively reduces to known ZO+subspace convergence analyses developed for ANNs, limiting the distinctiveness of the theoretical contribution.
- Practical advantage over strong baselines: After the inclusion of BP with gradient checkpointing and mixed precision, the claimed memory and efficiency advantages of SZO become highly regime-dependent. In realistic and commonly used settings (e.g., small time steps for image classification), BP-based methods can match or outperform SZO in memory usage, weakening the central argument that forward-only ZO provides a clear practical edge.
- Reliance on speculative neuromorphic deployment: Much of the paper’s motivation depends on future neuromorphic hardware where backward paths are infeasible. Several reviewers noted that such platforms remain largely experimental, and that the demonstrated advantages do not yet clearly translate into practical superiority on existing edge devices.

**Reviewer Scores:**

Reviewer 1k7D: Maintains a reject (2) stance, citing fundamental concerns about theoretical validity, novelty relative to existing ZO+subspace methods, and insufficient evidence of practical advantage. (but raised the score to 6 during the original rebuttal)

Reviewer nGip: Reject (2); review appears **unrelated**, but even excluding this review does not materially change the overall balance.

Reviewer fmgm: Borderline accept (6), explicitly stating they “would not mind if the paper is rejected.”

Reviewer GrS7: Borderline accept (6), satisfied after rebuttal but not strongly advocating acceptance.

---

### Decision · Program_Chairs · 2026-01-26

Reject